

# New particle formation induced by anthropogenic-biogenic interactions in the southeastern Tibetan Plateau

Shiyi Lai[1], Ximeng Qi[1, 2], Xin Huang[1, 2], Sijia Lou[1, 2], Xuguang Chi[1, 2], Liangduo Chen[1], Chong Liu[1],
Yuliang Liu[1, 2], Chao Yan[1, 2], Mengmeng Li[1], Tengyu Liu[1, 2], Wei Nie[1, 2], Veli-Matti Kerminen[3], Tuukka
Petäjä[3], Markku Kulmala[3] and Aijun Ding[1, 2]

[1]Joint International Research Laboratory of Atmospheric and Earth System Sciences, School of Atmospheric Sciences, Nanjing University, Nanjing, China
[2]Collaborative Innovation Center for Climate Change, Nanjing, Jiangsu Province, China
[3]Institute for Atmospheric and Earth System Research, Faculty of Science, University of Helsinki, Helsinki, Finland

*Correspondence to*: Ximeng Qi (qiximeng@nju.edu.cn) and Xin Huang (xinhuang@nju.edu.cn)

**Abstract.** New particle formation (NPF) plays a crucial role in the atmospheric aerosol population and has significant implications on climate dynamics, particularly in climate-sensitive zone such as the Tibetan Plateau (TP). However, our understanding of NPF in the TP is still limited due to a lack of comprehensive measurements and verified model simulations. To fill this knowledge gap, we conducted an integrated study combining comprehensive field measurements and chemical transport modeling to investigate NPF events in the southeastern TP during the pre-monsoon season. NPF was observed to occur frequently on clear-sky days in the southeastern TP, contributing significantly to the cloud condensation nuclei (CCN) budget in this region. The observational evidence suggests that highly oxygenated organic molecules (HOMs) from monoterpene oxidation participate in the nucleation in southeastern TP. After updating the monoterpene oxidation chemistry and nucleation schemes in the meteorology-chemistry model, the model well reproduces observed NPF and reveals an extensive occurrence of NPF across the southeastern TP. The dominant nucleation mechanism is the synergistic nucleation of sulfuric acid, ammonia and HOMs, driven by the transport of anthropogenic precursors from South Asia and the presence of abundant biogenic gases. By investigating the vertical distribution of NPF, we find a significant influence of vertical transport in the southeastern TP. More specifically, strong nucleation near the surface leads to an intense formation of small particles, which are subsequently transported upward.





These particles experience enhanced growth to larger sizes in the upper planetary boundary layer (PBL)

due to favorable conditions such as lower temperatures and reduced condensation sink. As the PBL evolves, the particles in larger sizes are brought back to the ground, resulting in a pronounced increase in near-surface particle concentrations. This study highlights the important roles of anthropogenic-biogenic interactions and meteorological dynamics in NPF in the southeastern TP.




## 1 Introduction

The Tibetan Plateau (TP) features the highest and most extensive highland in the world. Referred to as the "Third Pole", it exerts substantial impacts on global atmospheric circulation and the Asian monsoon climate (Qiu, 2008; Duan and Wu, 2005; Yanai et al., 1992; Liu et al., 2007; Liu et al., 2023). The thermal
forcing from the TP during spring contributes to the seasonal transition of East Asian circulation, playing a crucial role in determining the onset site and timing of the Asian summer monsoon (Liu et al., 2007; Yanai et al., 1992; Hsu and Liu, 2003; Sato and Kimura, 2007; Liang et al., 2005; Wu and Zhang, 1998). Additionally, the TP, serving as the "Asian water tower", is widely recognized as one of the most ecologically important areas (Lau et al., 2006; Xu et al., 2009; Lau et al., 2010; Immerzeel et al., 2010).
The substantial climate warming trend has been occurring over the TP since the last half century, altering the atmospheric circulation over half the planet and water supply for billions of people  (Niu et al., 2004; Yao et al., 2019; Chen et al., 2015; Lau et al., 2010; Immerzeel et al., 2010; Qiu, 2008). Of all the possible causes, clouds exert an important role in rapid warming in the TP, as they influence radiative balance and latent heating or cooling resulting from water phase changes (Ramanathan et al., 1989; Duan and Wu,
2006; Kang et al., 2010).

New particle formation (NPF) through gas-particle conversion is a vital contribution to cloud condensation nuclei (CCN) and thus significantly affects the cloud properties, especially in pristine areas (Merikanto et al., 2009; Wang and Penner, 2009; Yu and Luo, 2009; Kerminen et al., 2018). In most environments, sulfuric acid ($H_2SO_4$) is believed to play a crucial role in NPF due to its low volatility
(Kulmala et al., 2013). However, it has been observed that $H_2SO_4$ alone cannot fully explain the observed formation rates in the lower atmosphere (Kirkby et al., 2011). The efficient clustering of $H_2SO_4$ with bases has been reported (Almeida et al., 2013). Nucleation processes involving $H_2SO_4$ and stabilizing species such as ammonia and amines are considered the dominant mechanism for NPF in urban and rural environments (Yao et al., 2018; Lai et al., 2022b; Yan et al., 2021). Additionally, highly oxygenated
organic molecules (HOMs), formed through the oxidation of biogenic volatile organic compounds (BVOCs), are recognized as important drivers for NPF, especially under pristine or preindustrial conditions (Gordon et al., 2016; Gordon et al., 2017). The nucleation of pure biogenic organic species in the absence of $H_2SO_4$ was observed in the Cosmics Leaving Outdoors Droplets (CLOUD) chamber





(Kirkby et al., 2016). This mechanism is further supported by the measurements at the high-altitude sites (Bianchi et al., 2021; Bianchi et al., 2016). Notably, in forested areas and chamber experiments with the presence of both $H_2SO_4$ and BVOCs, $H_2SO_4$ clusters containing both base molecules and oxygenated organic molecules have been observed, suggesting the importance of multicomponent acid-base-organic nucleation ($H_2SO_4$-$NH_3$-Organics-$H_2O$) (Lehtipalo et al., 2018). Moreover, higher concentrations of ions at high altitudes could assist the above mechanisms (Kirkby et al., 2016; Yu, 2010; Frege et al., 2017). In general, various nucleation mechanisms, which are highly sensitive to the concentrations of precursor vapors and air ions, make the NPF complicated in the ambient.

Given the TP's pristine nature and limited anthropogenic activities, NPF is the main source of atmospheric aerosols and CCN (Carslaw et al., 2013; Gordon et al., 2017). Yet, our knowledge of NPF over the TP, especially the mechanisms of nucleation, is hampered due to the lack of comprehensive observations (Ma et al., 2008). A few ambient observations of NPF have been conducted at high-altitude sites over the TP. Neitola et al. (2011) showed that NPF takes place frequently in the pre-monsoon season at Mukteshwar (2180 m a.s.l.) in the western Himalaya. This is attributed to the lifting of the boundary layer height, allowing the transport of precursor gases, such as $SO_2$, from polluted lower-altitude regions to the station. Similarly, Moorthy et al. (2011) found a high probability of new ultrafine particle formation from precursor gases, potentially advected from inhabited regions, when solar insolation was abundant at the Hanle site (4520 m a.s.l.) in the Trans-Himalaya. A previous study conducted at the Himalayan Nepal Climate Observatory, Pyramid (NCO-P, 5079 m a.s.l.), located in the Eastern Himalaya, revealed that NPF events occurred frequently when more polluted air from valleys reached the site (Venzac et al., 2008). Recent comprehensive measurements at the NCO-P found a high frequency of NPF during cleaner conditions in the absence of $H_2SO_4$, suggesting that aerosol production in this region mainly occurs through organic precursors of biogenic origin (Bianchi et al., 2021).

Serving as a moisture transport bridge between the South Asian monsoon and the East Asian monsoon, the southeastern TP (average altitude is about 4200 m a.s.l.) is a climate-sensitive zone for modulating atmospheric circulation (Li et al., 2016). Notably, the southeastern TP is situated at the convergence of significant natural and anthropogenic aerosol sources, including densely populated areas to the south and extensive alpine forests on a regional scale. Substantial BVOCs from alpine forests could undergo fast



oxidation over the TP (Lin et al., 2008), favorable for the generation of HOMs. In addition to the natural process, anthropogenic emissions from the surrounding South Asia regions can also affect the TP through transport, contributing to the NPF-relevant precursors in the atmosphere (Zhao et al., 2013; Xia et al.,

2011). The interaction between biogenic and anthropogenic sources adds further complication as well as interest to the NPF in the southeastern TP. Model simulations constrained by available in situ measurements offer a promising approach to gaining a deeper understanding of the NPF process. To shed more light on NPF in the southeastern TP, we conducted an investigation focusing on NPF events during the pre-monsoon period, which is of paramount importance from the point of view of clouds and the

monsoon circulation. By integrating comprehensive in-situ measurements in Lulang and updated regional chemical model simulations incorporating process diagnoses, we aim to provide a picture of NPF from a three-dimensional perspective over the southeastern TP.

## 2 Materials and method

### 2.1 Measurement site and in-situ instrumentations

In this study, intensive in-situ measurements were conducted from 4 April to 24 May, 2021 in the Southeast Tibet Observation and Research Station (29°46' N, 94°44' E, 3200 m a.s.l.), a remote high-altitude site in Lulang of the southeastern TP.  During this pre-monsoon season, this region is under the control of the southern branch of the westerlies with the prominent southerly wind (Fig. 1a). In general,

the measurement site is characterized by extensive emissions of BVOCs, especially monoterpenes (Luo et al., 2002; Liang et al., 2011; Wang et al., 2007) (Fig. 1a). Despite a nearby cottage that emits pollutants from local burning during fixed hours in the morning, anthropogenic emissions in Lulang were relatively low on a regional scale (Fig. 1b).



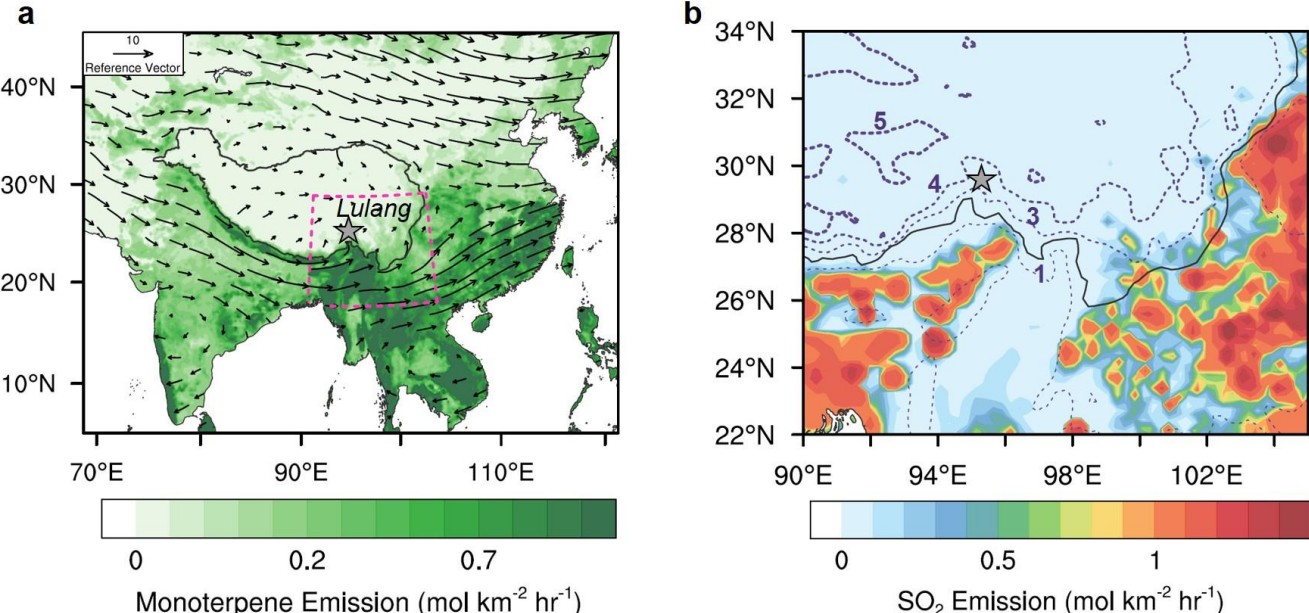

**Figure 1.** Spatial distribution of (a) biogenic monoterpene emission rates together with 700 hPa wind fields in March-April-May and (b) anthropogenic emission of $SO_2$ together with topographic field (purple isolines with the unit of km). Note: The black star marks the location of the Lulang site. The black solid line is the boundary of Tibetan Plateau. The pink dashed lines in Fig. 1a define the domain of Fig. 1b.

The particle number size distributions in the size range from 1 nm to 20 µm were collectively measured using five instruments, including a Particle Size Magnifier (PSM, Airmodus Inc.), a Neutral cluster and Air Ion Spectrometer (NAIS, Ariel Inc.), two Scanning Mobility Particle Sizers (nano-SMPS and long-SMPS, TSI Inc.) and an Aerodynamic Particle Sizer (APS, TSI Inc.). The PSM measures the size distribution of particles in the size range from 1nm to 3 nm (Vanhanen et al., 2011). The NAIS detects the air ion number size distribution from 0.8 nm to 40 nm and particle number size distribution from 2 nm to 40 nm (Mirme and Mirme, 2013). The nano-SMPS and long-SMPS measure particle size distributions over the size range of 4–70 nm and 12–540 nm, respectively. The APS measures the particle size distribution from 500 nm to 20 µm in the aerodynamic diameter. The monoterpene concentration was measured by a Proton Transfer Reaction Time-Of-Flight Mass Spectrometer (PTR-TOF-MS, Ionicon Analytik Inc.). A nitrate Chemical Ionization with the Atmospheric Pressure interface Time-Of-Flight mass spectrometer (CI-APi-TOF, Aerodyne Research Inc.) was used to detect the $H_2SO_4$ and HOMs



(Jokinen et al., 2012). Besides, on a typical NPF day, such as 29 April 2021, the CI-APi-TOF was switched to the APi-TOF negative mode to detect the chemical composition of atmospheric cluster ions (Junninen et al., 2010). The $PM_{2.5}$ concentration was observed using the online analyzer (SHARP5030,

Thermo Fisher Scientific Inc.). The Multi Angle Absorption Photometer (MAAP, Thermo-Scientific Inc.) was utilized to measure the black carbon (BC) mass concentration. The $SO_2$ and $O_3$ concentrations were observed using the API T100 and T400 (Teledyne API Inc.), respectively. The meteorological parameters, such as air temperature, wind speed and directions, were measured by meteorological sensors (WXT530, Vaisala Inc.).

## 2.2 WRF-Chem model configuration

To quantitatively understand the NPF process in the southeastern TP, the Weather Research and Forecasting model coupled with Chemistry (WRF-Chem) model version 3.9 is employed in this work. WRF-Chem is a widely used chemical transport model, which considers a variety of online-coupled meteorological processes and chemical processes, such as the emission and deposition of pollutants,

advection and diffusion, gaseous and aqueous chemical transformation, aerosol chemistry and dynamics (Grell et al., 2005). In this study, the simulation domain covers the southeastern TP, with a grid resolution of 27 km and 30 vertical layers from the ground level to the top pressure of 100 hPa. The simulation was conducted for 24–30 April, while the results of 28–30 April were analyzed to allow for the spin-up for the chemical initial condition. The initial and boundary conditions of meteorological fields were

constrained by the 6-hourly 1°×1° National Centres for Environmental Prediction (NCEP) global final analysis (FNL) data. Besides, NCEP Automated Data Processing (ADP) operation global surface observation and global upper air observational weather data was assimilated to improve the simulation of meteorological fields. The chemical initial and boundary conditions are provided by the whole atmosphere community climate model (WACCM) (Marsh et al., 2013). The WRF-Chem model

simulations consider both natural and anthropogenic emissions. The anthropogenic emissions, including those from power plants, residential combustion, industrial processes, on-road mobile sources, and agricultural activities, were obtained from the MIX Asian emission inventory database, with specific emissions for China sourced from the Multi-resolution Emission Inventory for China (MEIC) (Li et al.,





2017). The biogenic emissions were calculated online by using the Model of Emissions of Gases and
Aerosols from Nature (MEGAN) that embedded in WRF-Chem (Guenther et al., 2006). This module
estimates the net emission rates of monoterpene, isoprene, and other BVOCs from terrestrial ecosystems
to the above-canopy atmosphere.

The key parameterization options applied in this study include the Yonsei University (YSU) boundary
layer scheme (Hong et al., 2006), the Monin-Obukhov surface layer scheme (Jimenez et al., 2012), the
unified Noah land surface scheme (Ek et al., 2003), the Morrison microphysics scheme (Morrison et al.,
2009), the Grell-Freitas cumulus parameterization scheme (Grell and Freitas, 2014), and the Rapid
Radiative Transfer Model for General Circulation Models (RRTMG) longwave and shortwave radiation
(Iacono et al., 2008). For simulating gas-phase chemistry, the Statewide Air Pollution Research Center
(SAPRC-99) mechanism developed by Carter (2000) was employed. Aerosol lifecycle processes were
represented by the updated Model for Simulating Aerosol Interactions and Chemistry (MOSAIC) module,
which utilizes a sectional framework with 20 logarithmically spaced bins ranging from 1 nm to 10 μm.
This sectional approach enables better representations of nucleation and particle growth processes
(Lupascu et al., 2015; Matsui et al., 2011; Lai et al., 2022a; Lai et al., 2022b). The condensed volatility
basis-set (VBS) mechanism proposed by Shrivastava et al. (2011) was used to represent the gas-particle
partitioning and multi-generational gas-phase oxidation of organic vapors, with additional updates for
biogenic secondary organic aerosol (SOA) formation (Schervish and Donahue, 2020; Shrivastava et al.,
2015).

We evaluated the performance of our model by comparing it with measurements collected in Lulang. The
results demonstrate that our model is capable of well reproducing meteorological conditions and NPF-
relevant pollutants, including $SO_2$ and $PM_{2.5}$ (Table 1). To investigate the role of NPF and the specific
role of organic NPF pathways, we conducted three parallel numerical experiments: (1) a simulation with
all NPF pathways turned off (NPF-off), (2) a simulation considering all NPF pathways (NPF-on), and (3)
a simulation including only inorganic NPF pathways (NPF_inorg).



**Table 1.** Statistical analysis of the simulated 2-meter temperature, 10-meter wind speed, SO$_2$ concentration and PM$_{2.5}$ concentration versus the observations in Lulang.

|  | MB* | RMSE* |
| --- | --- | --- |
| 2-meter air temperature (ºC) | 1.07 | 2.3 |
| 10-meter wind speed (m s$^{-1}$) | 0.94 | 2.01 |
| SO$_2$ (ppb) | -0.14 | 0.16 |
| PM$_{2.5}$ (μg m$^{-3}$) | 0.52 | 7.32 |

* MB and RMSE refer to mean bias and root mean square error respectively.

## 2.3 Improvements of the new particle formation scheme in WRF-Chem

In this study, we employed a modified version of the VBS approach to calculate the organics contributing to NPF (Schervish and Donahue, 2020; Shrivastava et al., 2015). The default VBS mechanism represents non-traditional SOA derived from the oxidation of semi- and intermediate-volatility primary gases by two volatility species with saturation mass concentration (C$^*$) values of 10$^{-2}$ and 10$^5$ μg m$^{-3}$ (at 298K and 1atm) while treating traditional SOA produced by oxidation of biogenic and anthropogenic VOCs with C$^*$ values of 10$^0$ μg m$^{-3}$. Previous studies have demonstrated that oxidation of monoterpene could generate large amounts of ultra- and extremely low-volatility organic compounds (ULVOCs and ELVOCs), defined as organics with C$^*$ values smaller than 3×10$^{-9}$ μg m$^{-3}$ and 3×10$^{-5}$ μg m$^{-3}$, respectively, which could efficiently contribute to nucleation and initial growth process (Ehn et al., 2014; Zhao et al., 2020). Importantly, the formation of ULVOCs and ELVOCs occurs through autoxidation and dimerization processes involving peroxy radicals (RO$_2$), which differ from the traditional oxidation reactions commonly considered in the models (Crounse et al., 2013). To accurately depict the formation of low-volatility organic compounds from monoterpene, we extended the VBS mechanism to explicitly incorporate the chemistry of RO$_2$, including autoxidation, dimerization processes, and other competing reaction pathways. The resulting stable molecules from the termination of RO$_2$ radicals were then distributed into the appropriate volatility bins (Schervish and Donahue, 2020). To account for the



involvement of biogenic precursors in NPF, we adjusted the volatility representation of biogenic species within the VBS mechanism. Specifically, we reduced the lowest-volatility bin in the mechanism from $10^0$ µg m$^{-3}$ to $10^{-9}$ µg m$^{-3}$, allowing for the representation of ULVOCs and ELVOCs. Additionally, we expanded the volatility distribution to include eight bins with $C^*$ values ranging from $10^{-9}$ µg m$^{-3}$ to $10^5$

µg m$^{-3}$, separated by two orders of magnitude, to adequately capture the wide volatility range of organic vapors.

To simulate NPF processes, we incorporated a comprehensive ensemble of eight NPF parameterizations based on laboratory measurements into the MOSAIC module, coupled with the improved VBS scheme described above. These parameterizations accounted for both inorganic and organic nucleation pathways,

including binary neutral and ion-induced NPF ($H_2SO_4$-$H_2O$), ternary neutral and ion-induced NPF ($H_2SO_4$-$NH_3$-$H_2O$), amine- $H_2SO_4$ NPF ($H_2SO_4$-amine-$H_2O$), as well as three organic pathways, including multicomponent organic NPF ($H_2SO_4$-$NH_3$-Organics-$H_2O$), pure-organic neutral and pure-organic ion-induced NPF. The inorganic nucleation pathways are based on Dunne et al. (2016), and have been described in detail in our previous study Lai et al. (2022b). As for the organics-involved nucleation,

parameterizations of $H_2SO_4$-$NH_3$-Organics-$H_2O$ (Lehtipalo et al., 2018) and pure organics (Kirkby et al., 2016) were adopted. Importantly, Lehtipalo et al. (2018) have demonstrated that out of all HOMs, only organic vapors with ultra-low or extreme-low volatility are able to participate in nucleation. Thus, $C^*=10^{-9}$ µg m$^{-3}$ and $C^*=10^{-7}$ µg m$^{-3}$ species from monoterpene were considered as the nucleating vapors in our model. It should be noted that, as for the pure organic NPF, the original parameterization utilized the

measured HOMs with a wide volatility range ($C^*=10^{-20}$ –$10^2$ µg m$^{-3}$) as input variables, whereas we utilized the $C^*=10^{-9}$ µg m$^{-3}$ and $C^*=10^{-7}$ µg m$^{-3}$ species. To account for the changes in the input variables and maintain consistency, we introduced a linear adjustment factor of 6, which was determined as the ratio of the HOMs with $C^*<10^{-7}$ µg m$^{-3}$ to total HOMs measured in the laboratory study (Kirkby et al., 2016). Besides, a temperature dependence function following Dunne et al. (2016) was introduced to

account for the temperature effects on $H_2SO_4$-$NH_3$-Organics-$H_2O$ nucleation.



## 3 Results and discussions

### 3.1 Observational evidence for the enhanced NPF by anthropogenic and biogenic interactions

As a relatively remote natural area, NPF could be one of the main sources of CCN in southeastern TP.
We employed the method proposed by Kulmala et al. (2016) to estimate the source of CCN (details about
the method can be found in the Supplement). The particles with a size larger than 50 nm were used as the
CCN proxy. Fig. 2a illustrates the determination of the semi-empirical scaling factor (S1), which is crucial
for estimating the relative contributions of primary and secondary sources to CCN. To evaluate the
sensitivity of the estimation to S1, six different criteria were used. Specifically, the scatter plot was
constrained such that 0.1%, 0.5%, 1%, 1.5%, 2%, or 5% of the data points fell below the line represented
by S1×[BC]. Previous studies using the same method have reported S1 values ranging from $3×10^6$ to
$10×10^6$ CCN per ng BC (Rodriguez and Cuevas, 2007; Fernandez-Camacho et al., 2010; Gonzalez et al.,
2011). However, our study found that the S1 values for CCN in Lulang ranged between $0.15×10^6$ and
$0.60×10^6$ per ng BC, which are considerably smaller than those reported earlier. Fig. 2b presents the
average fraction of secondary particles in the CCN under different constraints for determining S1,
revealing variations between 62% and 90%. Despite the associated uncertainty, our findings suggest that
secondary sources, such as NPF, are the dominant contributors to the CCN budget in Lulang.

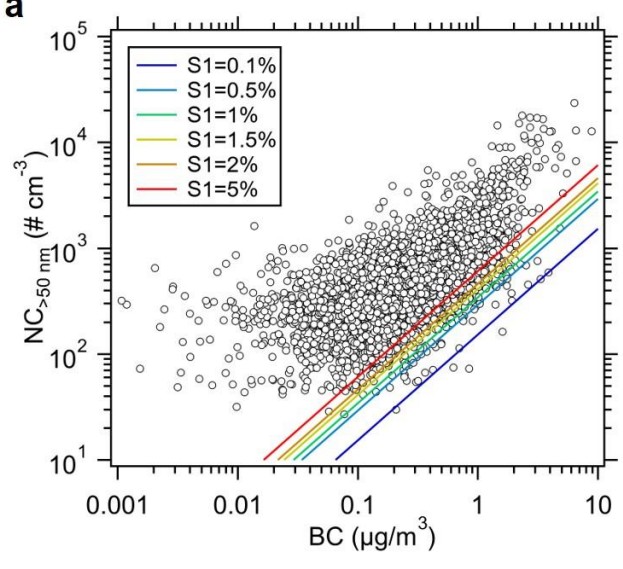
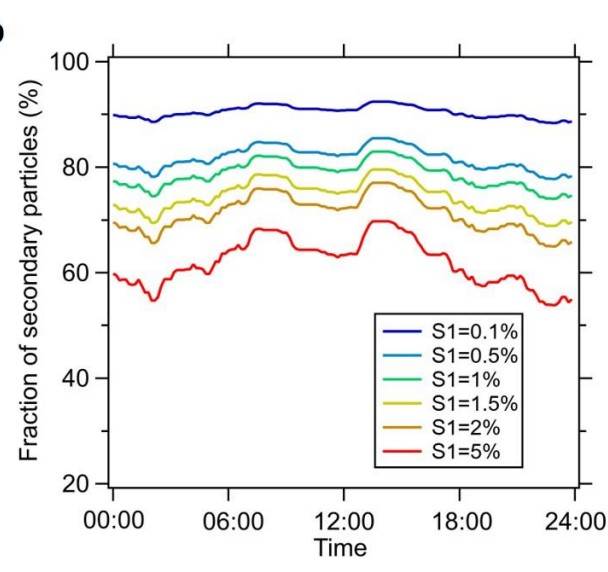



**Figure 2.** (a) Number concentration of particles larger than 50 nm (NC$_{>50\,nm}$) as a function of black carbon concentration [BC] in Lulang during the campaign. The lines represent those fittings for S1 in which 0.1%, 0.5%, 1%, 1.5%, 2% or 5% of the data points are located below the line represented by S1×[BC]. (b) Diurnal variation of the fraction of secondary source contributions to NC$_{>50\,nm}$ in Lulang during the campaign.

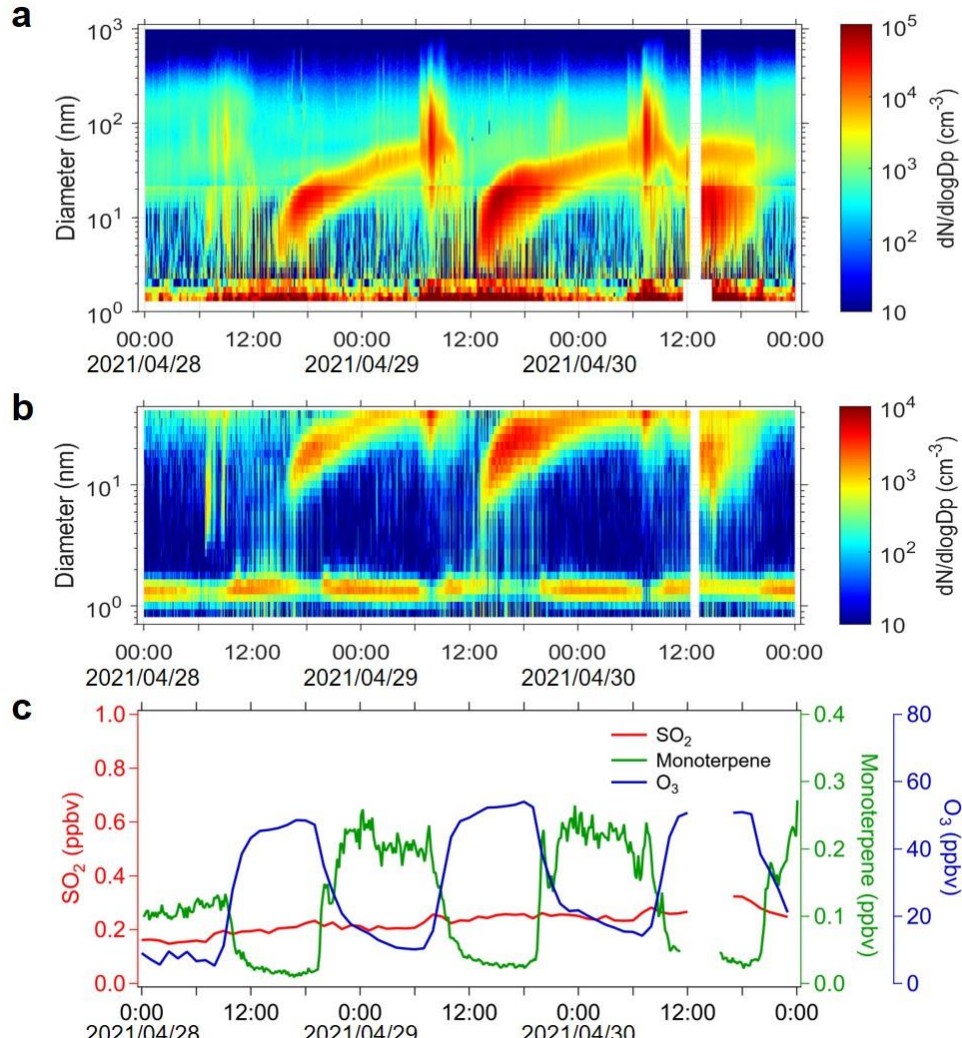

**Figure 3.** Temporal evolution of (a) particle number size distribution from 1 nm to 1000 nm, (b) positive ion number size distribution from 0.8 nm to 40 nm. Note that the high peaks of number concentrations around 10–200 nm observed in the morning was caused by the wood burning in a residential cottage near the site. (c) SO$_2$, monoterpene and O$_3$ concentrations measured in Lulang site during 28–30 April, 2021. All time is in the UTC+8 time zone in this study.






During our campaign, more than 60% of clear sky days were NPF event days, according to the criteria proposed by Dal Maso et al. (2005). Fig. 3a and Fig. 3b display the temporal variations of particle and ion number size distributions during a clear-sky episode from 28 April to 30 April 2021, respectively. Throughout this episode, typical NPF events were observed on a daily basis. Fig. 3c illustrates the

concentrations of NPF-relevant trace gases. The concentration of $SO_2$ in Lulang remained below 0.3 ppb, potentially approaching the lower detection limits. $O_3$ concentration displayed pronounced diurnal variation, with maximum concentrations exceeding 50 ppb during the daytime (12:00–18:00, UTC+8), indicating active photochemical reactions and a strong atmospheric oxidizing capacity. Due to the faster reactions with oxidants and enhanced boundary-layer mixing in the daytime, monoterpene, on the other

hand, showed a diurnal variation with higher mixing ratios at night and lower concentrations during the day. Specifically, the daily mean concentration of monoterpene was 118 ppt, with daytime mixing ratios ranging from 20 to 200 ppt and nighttime mixing ratios exceeding 0.26 ppb. This monoterpene concentration is comparable to the measurements in a boreal forest Southern Finland (Hyytiälä), where the average daily mixing ratio was 61 ppt during the spring season (Hakola et al. , 2012).

In Fig. 4a, the measured nucleation rates in Lulang are compared with laboratory experiments in the CLOUD chamber (Kürten et al., 2016; Lehtipalo et al., 2018) and ambient observations at the boreal forest in Hyytiälä (Sihto et al., 2006; Kulmala et al., 2013). The HOMs produced by the monoterpene oxidation have been demonstrated to play an important role in NPF at the boreal forest (Qi et al., 2018; Riccobono et al., 2014). As shown in Fig. 4a, the nucleation rate and $H_2SO_4$ concentration in Lulang are

similar with the observations in Hyytiälä, indicating a similarity in the nucleation process between the two locations. The nucleation exhibits a weak $H_2SO_4$ dependency and cannot be explained by the binary and ternary nucleation mechanisms involving $H_2SO_4$-$H_2O$ and $NH_3$-$H_2SO_4$-$H_2O$. However, the ambient atmospheric conditions in Lulang can be reproduced by the experiments considering three precursors (monoterpene, $H_2SO_4$, and $NH_3$), suggesting the significant role of the multicomponent acid-base-organic

nucleation mechanism ($H_2SO_4$-$NH_3$-Organics-$H_2O$). The interactions between precursors from anthropogenic sources and HOMs produced from biogenic VOCs could dominate the NPF process in Lulang. As shown in Fig. 4b, both $H_2SO_4$ and HOMs concentrations in Lulang exhibit pronounced diurnal





cycles, with peaks in the afternoon due to higher oxidation capacity during the daytime. The median daytime concentrations of $H_2SO_4$ and HOM concentration are about $10^6$ cm$^{-3}$ and $2\times10^8$ cm$^{-3}$, respectively,

consistent with measurements conducted in other forest environments (Riccobono et al., 2014). Figure 4c shows mass defect plots for negative ion clusters measured by APi-TOF during NPF on 29 April. The presence of organic compounds with 10, 15, or even 20 carbon atoms furtherly indicates the role of HOMs in nucleation.



**Figure 4.** (a) Nucleation rates ($J_{1.7}$) as a function of $H_2SO_4$ concentration at ambient observations in Lulang (green circles), Hyytiälä (gray circles) (Sihto et al., 2006; Kulmala et al., 2013) and CLOUD experiments (red diamonds) (Lehtipalo et al., 2018). The cyan and blue lines denote ternary ($H_2SO_4$-$NH_3$-



$H_2O$) nucleation and binary nucleation ($H_2SO_4$- $H_2O$), respectively, based on CLOUD data in Kürten et al. (2016). (b) Averaged diurnal variations of $H_2SO_4$ concentrations and HOMs concentration on NPF days in Lulang. The solid lines are the median values and shaded areas denote the 25th or 75th percentiles. (c) A mass defect plot illustrating the chemical composition of negative ion clusters at 12:00 on 29 April. The symbol size corresponds to the relative signal intensity on a logarithmic scale. (d) Formation rate at 10 nm ($J_{10}$) versus formation rate at 3 nm ($J_3$) at ambient observations in Lulang (diamonds). Diamonds are color-coded by condensation sink. Error bars present the 25th - 75th percentiles. The solid grey line shows the relationship between $J_{10}$ and $J_3$ based on theory (Kulmala et al., 2012) and the uncertainties are shown by the shaded bands. Dash 1:1 line is shown for reference.

In Fig. 4d, we compared the observed particle formation rates at 3 nm and 10 nm in Lulang, which are calculated based on NAIS measurement. According to the classical theory that neglects the influence of transport, it is expected that $J_{10}$ would be much lower than $J_3$ due to particle losses through coagulation scavenging as particles grow larger (Kerminen et al., 2001; McMurry et al., 2005; Kulmala et al., 2012). However, the observed $J_{10}$ is higher than what is predicted by the theory on most NPF days and notably higher than the observed $J_3$ on 29–30 April and 21–22 May. One possible explanation is that the larger particles observed were formed elsewhere and transported to the measurement site. A previous study also reported the significant role of transport, whether horizontally or vertically from upwind regions, in the NPF events in Lulang (Cai et al., 2018). The potential role of transport is also suggested by Fig. 3a, which shows a high number concentration of particles larger than 10 nm and relatively low concentrations of sub-10 nm particles, especially on 29 April. These findings suggest a non-negligible influence of transport on the observed NPF event.

## 3.2 Simulations on $H_2SO_4$-$NH_3$-Organics-$H_2O$ nucleation in the southeastern TP

To further quantify the roles of BVOCs and transport in the NPF in Lulang, we conducted regional simulations using the updated WRF-Chem regional atmospheric transport model for the typical NPF episode (28 April to 30 April 2021). Fig. 5a demonstrates that the model, incorporating a comprehensive ensemble of NPF pathways, was able to reproduce the occurrence of NPF events on these days. In addition, we compared the simulated concentrations of key NPF precursors with observations (Fig. 5b). The results showed that the model was able to reproduce the magnitude of the precursor concentrations, although it tended to underestimate the monoterpene concentration with a normalized mean bias of -16%. This could





be attributed to the simplification of vegetation classification and the limited understanding of the emission factors in the numerical descriptions (Guenther, 2013). To validate our simulation results, we

investigated the behavior of particle number concentrations in the range of 1–10 nm ($CN_{1-10}$) and 10–40 nm ($CN_{10-40}$). As shown in Fig. 5c, the simulated $CN_{1-10}$ and $CN_{10-40}$ were quite close to observations, which is consistent with previous NPF simulation studies (Lupascu et al., 2015; Lai et al., 2022a). We compared the simulated particle number size distributions with the observations (Fig. 5d). The inclusion of inorganic NPF ("NPF_inorg") narrowed the gap between simulation and observation, but still could

not reproduce the observed peak of small particles (<40 nm diameter). The further incorporation of organic NPF pathways (NPFon) bridged the remaining gap and resulted in a fairly good agreement with observations, indicating the vital role of organics in nucleation.

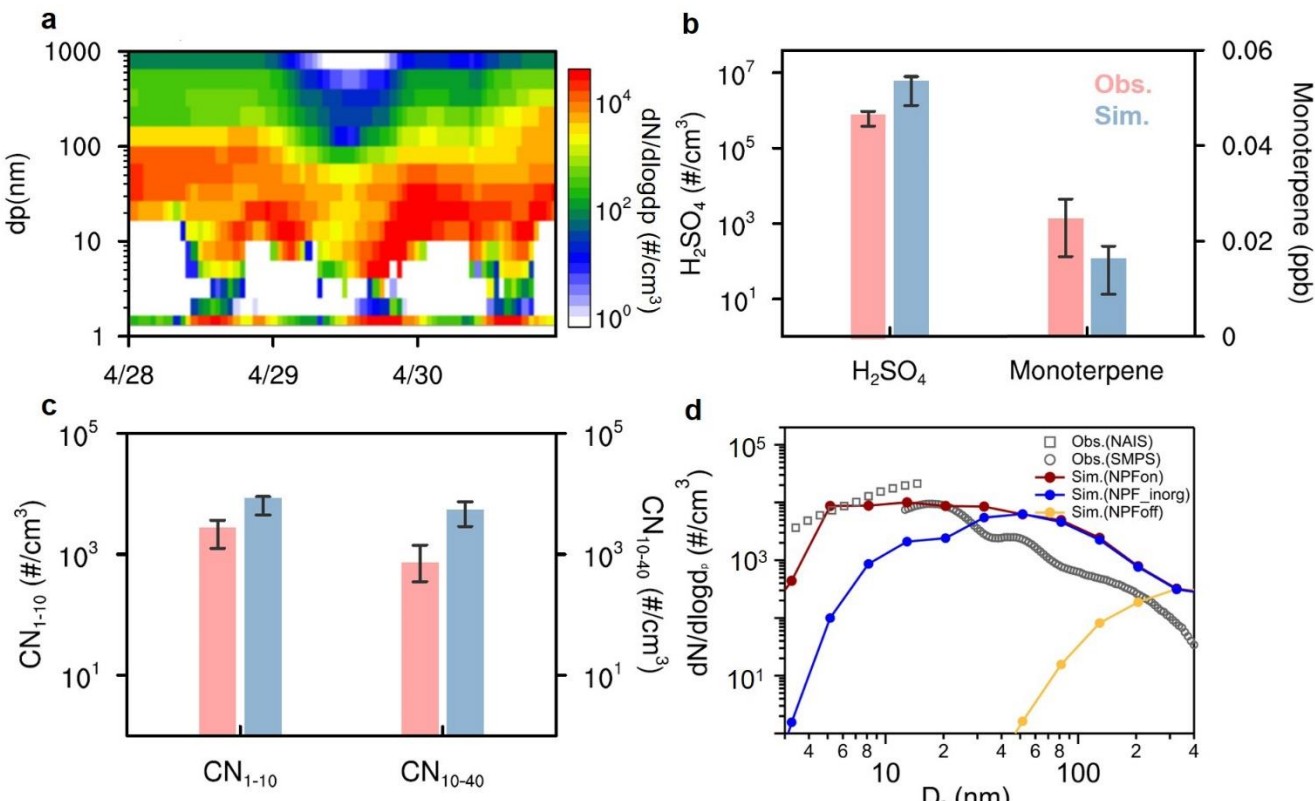

**Figure 5.** (a) Simulated particle number size distribution in Lulang. (b) Observed and simulated $H_2SO_4$

and monoterpene concentration in Lulang. Bars are the median value during 28–30 April. The whiskers represent the 25–75th percentile. (c) Same as Fig. 5b but for number concentrations of 1–10 nm and 10–40 nm particles. (d) The observed and simulated particle number size distributions averaged from 12:00



to 18:00 during 28–30 April. The gray squares and gray circles show the measurements by NAIS and SMPS, respectively. Note: the deviations between the two sets of instruments were attributed to

systematic bias due to measurement principle. Red, blue and orange lines represent the NPF-on, NPF_inorg and NPF-off experiments.

The typical "banana" shape particle number size distribution showing nucleation and subsequent growth throughout the day at the Lulang site (Fig. 3a) suggests that nucleation and growth occur on a regional

scale. Fig. 6a presents a simulated spatial distribution of near-surface $CN_{1-10}$ averaged during the episode, revealing high concentrations of $CN_{1-10}$ in southeastern TP. Our model results indicate that the nucleation and growth of aerosol particles extended over a large regional area, covering several thousand kilometers in southeastern TP. To gain a deeper understanding of the nucleation and growth process in southeastern TP, we extend the analysis from Lulang to the NPF-intensive surrounding area, as marked by the pink

dashed rectangle in Fig. 6a. Fig. 6b illustrates the relative contribution of different NPF pathways within this area. Our findings indicate that the pathways involving organic compounds dominate the NPF rates in the southeastern TP, with $H_2SO_4$-$NH_3$-Organics-$H_2O$ nucleation and pure-organic ion-induced NPF accounting for 85.0% and 12.6%, respectively.

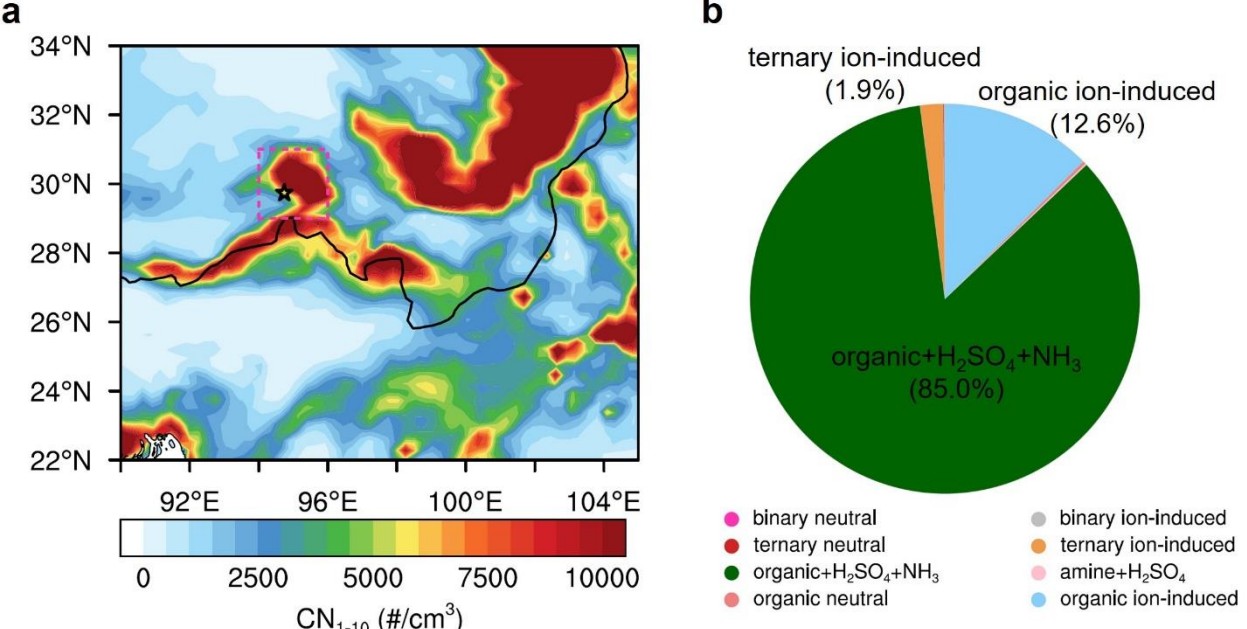




**Figure 6.** (a) Averaged spatial distributions of simulated near-surface 1–10 nm particle number concentrations during 28–30 April. Note: The black star marks the location of the Lulang site. The pink dashed rectangle defines further research domain. (b) The relative contribution of different NPF pathways to near-surface nucleation rate averaged over region indicated by the pink dashed rectangle in Fig. 6a during 28–30 April.

**Figure 7.** Averaged spatial distributions of simulated near-surface (a) $SO_2$ concentration, (b) OH concentration, (c) condensation sink, (d) $H_2SO_4$ concentration during 28–30 April. Note: The black star marks the location of the Lulang site.





To investigate the sources of gaseous precursors of $H_2SO_4$-$NH_3$-Organics-$H_2O$ nucleation in the southeastern TP, we present simulated spatial distributions of $SO_2$, OH, $H_2SO_4$ concentration, and condensation sink (CS) at the ground surface during the NPF episode (Fig. 7). The low anthropogenic emissions in the region lead to an average $SO_2$ concentration of less than 0.5 ppb, which is supported by our observations (Fig. 7a). In contrast, the OH radical concentration, which is a key oxidant for forming

$H_2SO_4$ and an indicator of atmospheric oxidizing capacity, is high in the southeastern TP (Fig. 7b). This high concentration is consistent with previous studies and can be attributed to strong solar radiation and a high ozone background over the TP (Ren et al., 1999; Lin et al., 2008; Yan and Yu, 1997; Zhu et al., 2006). Additionally, the abundant water vapor in the southeastern TP further enhances the chemical production of OH radicals (Wu et al., 2005; Wang et al., 2015). CS, which quantifies the role of pre-

existing particles in removing condensable vapors and newly formed particles from the atmosphere, is expected to suppress NPF (Kulmala et al., 2012; Kerminen et al., 2018; Qi et al., 2019). Due to the relatively pristine environment, lower CS values are found over the TP compared with those in South Asia (Fig. 7c). As a result of strong atmospheric oxidizing capacity and low CS, $H_2SO_4$ concentrations can reach certain values despite the low $SO_2$ concentration and are higher on the TP compared to the

surrounding area.

Ammonia ($NH_3$) is also a significant precursor for particle formation, mainly emitted from agricultural, domestic, and industrial activities (Behera et al., 2013). Given the large footprint of agriculture in this region, the South Asia region, located to the south of the TP, has been identified as a hotspot for ammonia emissions (Clarisse et al., 2009; Van Damme et al., 2018; Xu et al., 2018). Fig. 8a shows that high levels

of near-surface $NH_3$ were simulated in South Asia, and the southerly wind transported $NH_3$ to the TP during the daytime. The vertical simulation of zonal averaged wind vector and $NH_3$ concentrations over 94–96 °E during the daytime (Fig. 8b) corroborates the results, showing an overall northward circulation that tends to advect $NH_3$ from South Asia to the TP. This transport contributes to considerable $NH_3$ concentrations and facilitates $H_2SO_4$-$NH_3$-Organics-$H_2O$ nucleation in the southeastern TP. Besides, the

southeastern TP is characterized by high monoterpene concentration due to the extensive coverage of evergreen forests and high monoterpene emission (Fig. 1a and Fig. 8c). Given the substantial





concentration of monoterpene and the strong oxidizing capacity in the region, abundant oxidized biogenic gases are formed in the southeastern TP (Fig. 8d), which could significantly contribute to the NPF (Ehn et al., 2014; Bianchi et al., 2016; Qi et al., 2018). In summary, NPF is promoted in the southeastern TP through anthropogenic-biogenic interactions.

**Figure 8.** (a) Spatial distributions of simulated near-surface NH$_3$ concentration together with surface wind fields averaged from 12:00 to 18:00 during 28–30 April. (b) Latitude–height cross sections of NH$_3$ concentrations together with wind fields averaged along 94–96°E from 12:00 to 18:00 during 28–30 April. Note: The pink dashed lines show the location of the research domain marked in Fig.6a. (c) Averaged



spatial distributions of simulated near-surface monoterpene concentration during 28–30 April. (d) Same as Fig. 8c but for oxidized biogenic gas. Note: The black star marks the location of the Lulang site.

## 3.3 The vertical heterogeneity of NPF in the southeastern TP

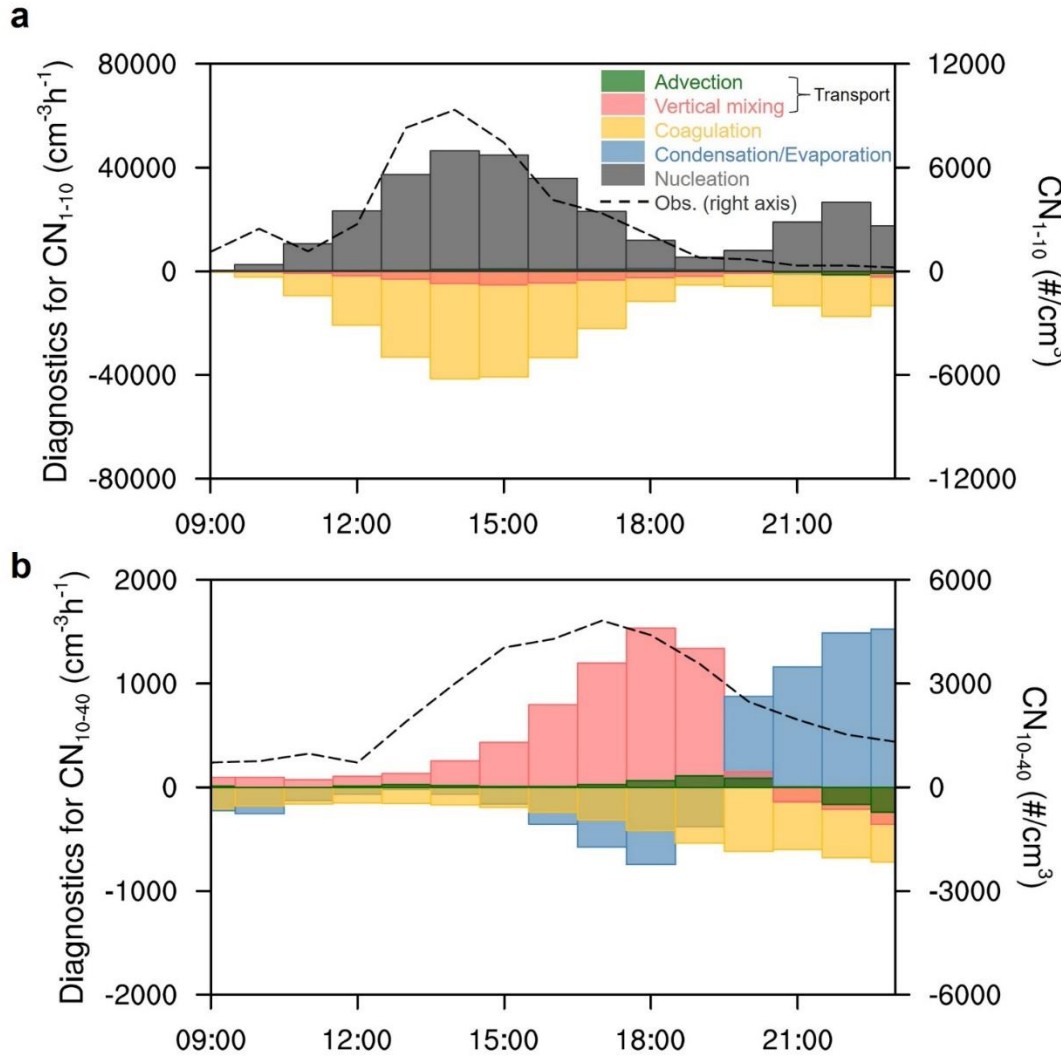

**Figure 9.** (a) Diurnal evolution of contributions of near-surface 1–10 nm particle number concentrations from different processes calculated by WRF-Chem processes analysis during 28–30 April over the research domain. The black dashed line represents the diurnal evolution of observed 1–10 nm particle number concentrations during 28–30 April. (b) Same as Fig. 9a but for 10–40 nm particle number concentrations.




The observational evidences presented in section 3.1 as well as Fig. 4d suggest that the observed NPF was highly possible to be influenced by regional transport and convective motions. To further investigate the impacts of different chemical and physical processes on NPF, we conducted the diagnostic analysis using the WRF-Chem model. Our focus was specifically on $CN_{1-10}$ and $CN_{10-40}$, which serve as indicators of $J_3$ and $J_{10}$, respectively. Fig. 9 illustrates the diurnal evolution of the relative contributions from nucleation, condensation/evaporation, coagulation, and transport (vertical mixing and advection) to near-surface $CN_{1-10}$ and $CN_{10-40}$, along with the observed $CN_{1-10}$ and $CN_{10-40}$. $CN_{1-10}$ exhibits a dominant peak around 14:00, which is closely associated with the nucleation process (Fig. 9a). The main loss of $CN_{1-10}$ is coagulation and vertical mixing. As for $CN_{10-40}$, a peak is observed around 17:00, with a time lag of 3 hours compared to $CN_{1-10}$. The diagnostic analysis reveals that vertical mixing is the dominant process that enhances the surface $CN_{10-40}$ during the daytime, with an estimated contribution of 272% (Fig. 9b). The condensation process shows a negative contribution to $CN_{10-40}$ during the daytime and a prominent positive contribution in the evening, indicating that the particles shrink and grow through evaporation and condensation, respectively. Similar to $CN_{1-10}$, coagulation serves as one of the main losses for $CN_{10-40}$. In summary, our diagnostic analysis reveals that nucleation predominantly drives the production of near-ground $CN_{1-10}$, while vertical mixing enhances the concentration of particles in the larger size range near the ground.

To gain further insights into the vertical distribution of the NPF process, we analyzed the simulated vertical profiles of nucleation rate during the NPF episode over the research domain (Fig. 10a). The nucleation rate exhibits a distinct vertical pattern, with peak values observed near the surface within the planetary boundary layer (PBL), as well as another peak at around 5 km in the free troposphere. This vertical distribution suggests that nucleation not only occurs predominantly in the PBL but also extends to higher altitudes in the atmosphere. We further examined the relative importance of each nucleation mechanism (Fig. 10b). At lower altitudes, $H_2SO_4$-$NH_3$-Organics-$H_2O$ nucleation and organic ion-induced nucleation dominate the nucleation rate, accounting for 75% and 12% below 2 km, respectively. As the altitude increases, $H_2SO_4$-$NH_3$-$H_2O$ nucleation becomes the primary pathway between 4 and 5 km, with ion-induced and neutral pathways contributing 12% and 83%, respectively. $NH_3$ exhibits higher concentrations near the surface within the PBL and another peak at around 5 km (Fig. S1a in the





Supplement). The higher concentrations of NH₃ at lower altitudes are attributed to transport from South
Asia by the near-surface southerly winds (Fig. 8a). Those NH₃ could then transport into the upper
troposphere through the convection over the TP. The presence of elevated NH₃ concentrations in the free
troposphere has also been reported in previous studies based on aircraft observations (Hopfner et al.,
2016; Höpfner et al., 2019).

**Figure 10.** (a) Averaged diurnal evolution of vertical cross section of simulated nucleation rate over the
research domain during 28–30 April. Note: The black dashed line shows the planetary boundary layer
height (PBLH). (b) The relative contribution of different NPF pathways averaged over the research
domain during 28–30 April. (c) Same as Fig. 10a but for 10–40 nm particle number concentrations. (d)
Same as Fig. 10a but for 10–40 nm biogenic organic mass concentrations.


After nucleation, the newly-formed particles undergo growth through condensation of vapors onto their surface, resulting in an increase in particle size. Interestingly, in contrast with the vertical distribution of nucleation rate, the $CN_{10-40}$ exhibited a notable counter-gradient during the daytime (Fig. 10c). This pattern is consistent with the distribution of biogenic organic mass concentration (Fig. 10d), indicating

the important role of biogenic organic vapors in this region. The lower $PM_{2.5}$ concentration at the boundary layer top suggests a lower condensation sink for condensable gases and a higher survival probability of freshly nucleated particles (Fig. S1b). Additionally, the lower temperature at higher altitudes creates more favorable conditions for the condensation process of organic vapors, facilitating the subsequent growth of nanoparticles (Fig. S1c).

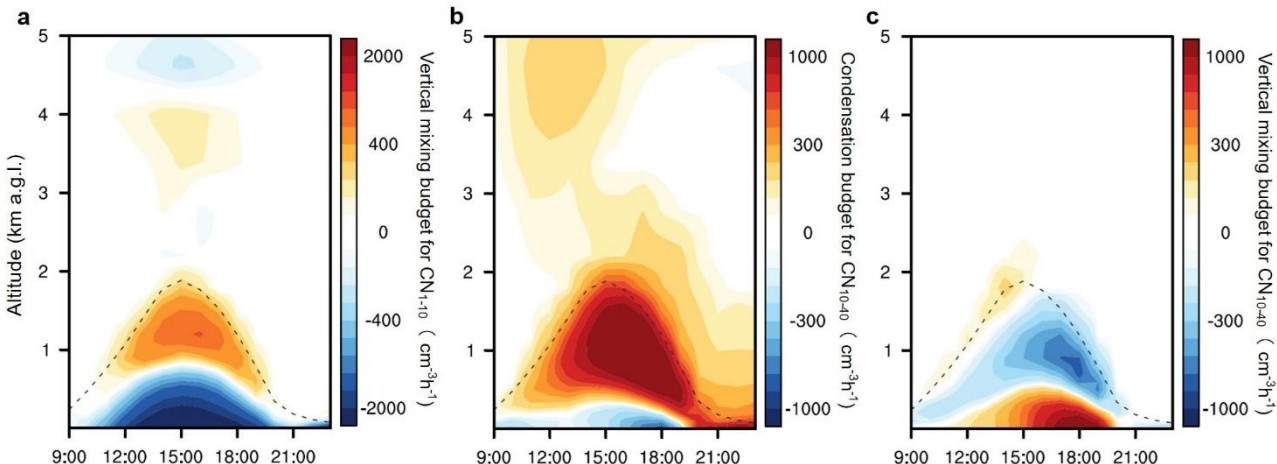


**Figure 11.** Averaged diurnal evolution of vertical cross section of (a) vertical mixing tendencies for 1–10 nm particle number concentrations, (b) condensation tendencies for 10–40 nm particle number concentrations, (c) vertical mixing tendencies for 10–40 nm particle number concentrations during 28–30 April over the research domain. Note: Black dashed line denotes PBLH.


Figure 11 further elucidates the key processes in shaping the vertical heterogeneity of NPF. As discussed in section 3.2, the near-surface nucleation induced by anthropogenic-biogenic interaction is facilitated over the southeastern TP, leading to the formation of substantial nanoparticles. These newly formed nanoparticles are then transported upwards through vertical mixing within the PBL (Fig. 11a). As these





particles ascend, they encounter a notable reduction in air temperature and $PM_{2.5}$ concentration in the upper PBL (Fig. S1b, c). These altered conditions create an environment conducive to the condensational growth of small particles (Fig. 11b), resulting in an increased concentration of particles within the 10–40 nm size range in the upper air (Fig. 10c). Subsequently, the particles that underwent growth through condensation in the upper air are effectively brought back down to lower altitudes by vertical mixing (Fig.

11c). This process contributes to a significant enhancement of particle concentration in the size range of 10–40 nm near the surface, as observed in the $CN_{10-40}$ measurements (Fig. 9b). These findings highlight the importance of vertical mixing in the vertical redistribution of particles and provide valuable insights into the complex dynamics of NPF in the southeastern TP.

## 4 Conclusions

Combining comprehensive in-situ measurements and an updated regional chemical transport model, this study provides a comprehensive investigation of the NPF occurring in the southeastern TP during the pre-monsoon period. The measurement site, Lulang, exhibits a regionally low anthropogenic emission but high biogenic emission environment, with trace-level concentrations of $SO_2$ and high concentrations of monoterpene. Our findings reveal the significant contribution of secondary sources to the CCN budget in

Lulang, highlighting the importance of NPF in this region. On clear-sky days, a high frequency of NPF exceeding 60% was observed. Observational evidence shows that the multicomponent organic ($H_2SO_4$-$NH_3$-Organics-$H_2O$) mechanism is the dominant nucleation in Lulang, and HOMs formed from monoterpene oxidation play an important role. Model results further confirm that this nucleation mechanism prevails in the whole southeastern TP and reveals that $NH_3$ was transported from South Asia.

Importantly, our study also unveils the crucial role of vertically heterogeneous NPF and vertical mixing in shaping the particle number size distribution near the surface. The vertical distribution of precursors (such as monoterpene and $NH_3$) leads to strong nucleation near the ground, and newly formed small particles are transported upward. The upper PBL exhibits lower temperature and $PM_{2.5}$, favorable for condensational growth, resulting in a counter-gradient pattern for $CN_{10-40}$ during the daytime. As the PBL

evolves, larger particles are transported to the ground, leading to a significant enhancement of $CN_{10-40}$

near the surface. These processes, including the anthropogenic-biogenic NPF and meteorological dynamics, can explain the observed high ratio of $J_{10}$ to $J_3$ during the NPF in southeastern TP.

This study gives valuable insights into the prevailing mechanism driving NPF in the southeastern TP, highlighting the interaction between anthropogenic and biogenic sources in NPF processes in this region.

We emphasize the importance of considering vertical mixing to improve our understanding of NPF and its implications for climate change. Additionally, our findings underscore the significant role of NPF at the high altitude, particularly in pristine regions like the TP, as they can profoundly influence cloud properties and have broader implications for climate dynamics.


**Data availability**

The anthropogenic emission data MIX is available at http://www.meicmodel.org/dataset-mix.html. The gridded anthropogenic emissions for China are available at http://meicmodel.org/?page_id=541&lang=en. The 700 hPa wind fields are obtained from the fifth-generation European Centre for Medium-Range

Weather Forecasts (ECMWF) reanalysis data (ERA5; https://cds.climate.copernicus.eu/cdsapp#!/home). The in-situ measurement data in southeastern Tibetan Plateau are available from the corresponding author upon request.

**Competing interests**

At least one of the (co-)authors is a member of the editorial board of Atmospheric Chemistry and Physics.

**Author contributions**




XQ, XH and AD conceptualized and supervised the study. XQ, XC, LC, CL, YL, TL and WN conducted the in-situ measurement. SL, XQ, XH, SL, YC and ML contributed to the model development and simulation. SL, XQ and XH analyzed the data and interpreted the results with inputs from SL and YC. SL wrote the original draft. SL, XQ and XH reviewed and edited the paper with contributions from all co-authors.


**Acknowledgemets**

This work was supported by the second Tibetan Plateau Scientific Expedition and Research (STEP) program (grant no. 2019QZKK0106) and the National Natural Science Foundation of China (41922038, 42175113).

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
