# Peer review of "New particle formation induced by anthropogenic-biogenic interactions in the southeastern Tibetan Plateau"

_EGUsphere, 2023_

## Referee Comment (RC2)

Review of New particle formation induced by anthropogenic-biogenic interactions in the southeastern Tibetan Plateau, Lai et al.

This manuscript contains a suite of high-quality measurement data showing $H_2SO_4$, HOMs, and a PNSD from the Tibetan plateau. Frequent NPF was observed and an analysis of both the measurement data and some very impressive WRF-Chem simulations are presented giving some fascinating insights into the chemistry on a larger scale than measurements allow. The WRF-Chem simulations are possible due to some improvements to the VBS. The methodology for both the measurements and model are, however, extremely sparse. The developments to VBS are not discussed at all, neither are the processing of the measurement data. Similarly, the analysis of the mass spectral data is not given enough time. Many of the key arguments depend on the HOMs being monoterpene oxidation products, but the mass spectra are not discussed in detail. I understand that this would result in a very large paper so I think these things belong in the supplement. Once these are addressed, I very highly recommend this for publication as it is an extremely impressive paper.

**General comments**

1) The methodology regarding instrumentation is a little bit thin-on-the-ground. It would be nice to have more information about the equipment. What flowrates were the instruments run at? Did they share an inlet? Did the PSM run in scanning mode? What were the time resolution? How were the PTR and CIMS instruments calibrated? What about mass-dependent transmission corrections? This information can go in the supplement, but it is important.

2) The improvements to WRF-Chem are very valuable! But barely discussed. You say you updated VBS to incorporate $RO_2$ chemistry, including autoxidation and dimerization! Is this similar to existing work such as PRAM/autoPRAM? (1). This should definitely be discussed in detail (again, even if only in the supplement) as the outcomes of the paper hinge on these results. Also, are there plans to make these improvements available widely?

3) Similar to the previous comments, as your arguments hinge on the HOMs being monoterpene oxidation products, it would be nice if you showed them in more detail. The mass defect shows them, but you've lumped $C_{6-10}$ together. Why not colour it by carbon number? I'd also like to see DBE per carbon, and average oxidation state. Otherwise I have no idea what the HOMs actually *are*. I'd need to be satisfied that the HOMs are similar to alpha-pinene oxidation products, as many of the later arguments depend on alpha pinene lab studies.

**Specific comments**

Line 223: I'm not sure I understand the logic here, why only those species with $C^*$ of $10^{-9}$ and $10^{-7}$? Why not $10^{-8}$? Surely this would avoid you having to input the factor-of-six adjustment? Also,

Line 229: Can you explain the temperature dependence function?

Line 279: Not sure Qi and Riccobono are the right references here. Maybe these two: (2, 3) as the former shows the formation of particles primarily through HOMs, while the latter shows the importance of HOMs as well as $H_2SO_4 + NH_3$ in the boreal environment a little more accurately than Riccobono. Maybe also (4) to show their role in growth.

Figure 3: This figure is great. Is it possible to include one for the whole campaign including $H_2SO_4$ and HOM? Also maybe use a different colour palette other than Jet (maybe Turbo or Viridis). Same for the other figures.

Figure 4c: As above, the sequence of greens is quite difficult to understand here. Also, is the choice of red + green for sulphuric acid + $H_2SO_4$ color blind friendly?

Figure 5b,c: I'm finding these bar charts slightly hard to read. Why do the charts start at $<10^1$ cm$^{-3}$ ? It makes the actual difference quite hard to see. Why not a boxplot with a Y axis? Then we'd be able to see the min/max concentrations measured & predicted by the model, as well as the distribution, and median value.

Figure 5d: It looks like there's a factor of 2-3 difference between the point where the NAIS and SMPS cross over. Do you have a reference instrument you can correct to? If not, it's common practice to correct the NanoSMPS/NAIS to the LongSMPS. In either case, it doesn't make any difference to the conclusions of the figure.

Line 384: Do you mean "high values" rather than "certain values"?

Line 386 (and following paragraph): What about the SO$_2$? If that is also anthropogenic (which I'd presume it is as I doubt there's much DMS up there) then this strengthens your biogenic-anthropogenic argument.

Figure 10: Is nucleation rate here J$_{1.5}$, J$_3$, or J$_{10}$? Also, it might be easier to read if instead of "binary" and "ternary" you put H$_2$SO$_4$-H$_2$O and H$_2$SO$_4$-NH$_3$-H$_2$O.

Line 461: What fraction of total CN$_{10-40}$ mass does this Biogenic organic mass comprise?

Line 462: Maybe worth considering that autoxidation rates also decrease with temperature (5)

1.      L. Pichelstorfer *et al.*, Towards a mechanistic description of autoxidation chemistry: from precursors to atmospheric implications. *EGUsphere* **2023**, 1-30 (2023).
2.      C. Rose *et al.*, Observations of biogenic ion-induced cluster formation in the atmosphere. *Science Advances* **4**, eaar5218.
3.      K. Lehtipalo *et al.*, Multicomponent new particle formation from sulfuric acid, ammonia, and biogenic vapors. *Science Advances* **4**, eaau5363.
4.      C. Mohr *et al.*, Molecular identification of organic vapors driving atmospheric nanoparticle growth. *Nature Communications* **10** (2019).
5.      D. Stolzenburg *et al.*, Rapid growth of organic aerosol nanoparticles over a wide tropospheric temperature range. *Proceedings of the National Academy of Sciences* **115**, 9122-9127 (2018).

---

## Author Comment (AC1)

**General comments:**

*New particle formation (NPF) is essential for understanding aerosol dynamics, particularly in climate-sensitive regions like the Tibetan Plateau (TP). To address the knowledge gaps specific to NPF in the TP, the authors conducted a study that combined field measurements and modeling, revealing frequent NPF events in the southeastern TP. Lulang, characterized by relatively low anthropogenic emissions but high biogenic emissions and trace-level $SO_2$ concentrations, along with high monoterpene concentrations, served as a pertinent study area.*

*To assess the influence of biogenic volatile organic compounds (BVOCs) and atmospheric transport on NPF in Lulang, a well-chosen set of regional simulations was performed using the updated WRF-Chem model during a typical NPF episode. For the most part, the conclusions were substantiated. The inclusion of organic NPF pathways helped bridge remaining gaps and yielded good agreement with observations, underscoring the vital role of organic compounds in nucleation. Crucially, oxygenated organic molecules resulting from monoterpene oxidation were identified as pivotal for NPF. The model updates successfully replicated observed NPF, highlighting the dominant nucleation mechanism involving sulfuric acid, ammonia, and highly oxygenated organic molecules. Vertical transport was found to have a substantial impact on NPF in the southeastern TP, emphasizing the significance of anthropogenic-biogenic interactions and meteorological dynamics. The analysis disclosed that nucleation primarily drives the production of near-ground particles below 10 nm range, while vertical mixing enhances particle concentration in larger size ranges near the ground. These findings underscore the importance of vertical mixing in redistributing particles within the southeastern TP.*

*In conclusion, this well-written and compelling paper provides valuable insights into the complex dynamics of NPF, focusing on the roles of different precursors and meteorological factors in the formation and growth of aerosol particles. However, I have some reservations about two aspects: the potential limitations of the 27 km grid resolution and the climatology of NPF events, specifically the model's ability to accurately represent the vertical transport of aerosols back into the boundary layer.*

**Response:** We would like to thank the referee for providing the insightful suggestions, which indeed help us further improve the manuscript.

**Specific comments:**

*Comment 1: Page 5, line 109-113 "(Fig. 1a). In general, the measurement site is characterized by extensive emissions of BVOCs, especially monoterpenes ... (Fig. 1b)."*

*Please provide more details regarding the sources, data level, and resolution depicted in Figures 1a and 1b. The mention of a "nearby cottage" prompts the need for clarification: is this a single cottage, or does it represent a local distribution of cottages surrounding the measurement site? This is particularly crucial in assessing the direct emissions originating from these cottages.*

**Response:** Thanks for the comment. The biogenic emissions in Figure 1a were calculated online by using the Model of Emissions of Gases and Aerosols from Nature (MEGAN) that embedded in WRF-Chem (Guenther et al., 2006). This module estimates the net emission rates of monoterpene, isoprene, and other BVOCs from terrestrial ecosystems to the above-canopy atmosphere. Anthropogenic emissions are obtained from the mosaic Asian anthropogenic emission inventory (MIX) at 0.25º×0.25º horizontal resolution and a monthly temporal resolution for the year 2010, with specific emissions for China sourced from the Multi-resolution Emission Inventory for China (MEIC) at 0.25º×0.25º horizontal resolution for 2017 (http://meicmodel.org.cn) (Li et al., 2017b). Anthropogenic emissions encompass those from power plants, residential combustion, industrial processes, on-road mobile sources, and agricultural activities (Li et al., 2017b). Figures 1a and 1b were plotted at the model resolution (27 km). In response to your valuable comments, we have included additional details in the Lines 175–177 in Section 2.2 and the captions of Figure 1 (Lines 122-126).

Regarding the biomass burning signals in the morning, there was a single residential cottage close to the site, which emitted pollutants by burning wood at almost regular morning hours. It is crucial to note that these biogenic burning signals represent highly localized emissions, which need to be excluded from data analysis. In line with your suggestion, we have clarified the role of the cottage in the Lines 113–114 in the revised Section 2.1 of the manuscript.

*Comment 2: Page 7, line 132-134 "Besides, on a typical NPF day, such as 29 April 2021, ..."*

*Please explain the climatology/frequency of NPF days and describe the nature of a typical NPF day. As there have been no references to NPF events thus far, providing some context regarding the occurrence of NPF events at the study location would be helpful.*

**Response:** Thanks for the comment. On typical NPF days, the contour plot depicting particle number size distributions over time at a fixed location often exhibits a "banana shape". The banana-shaped NPF event signifies the regional occurrence of this phenomenon, spanning a minimum radius of a few tens of kilometers (Kulmala et al., 2012). We applied a methodology proposed by Dal Maso et al. (2005) to identify NPF-event days during our 47-day

campaign, revealing a total of 10 NPF days. This frequency is slightly lower than observations during the monsoon season, where over 80% (31/38 days) of days were reported to exhibit NPF by Cai et al. (2018). The relatively moderate frequency during our campaign could be attributed to factors such as high cloud coverage, frequent rain, and low radiation during the pre-monsoon period. Notably, on clear sky days, more than 60% were identified as NPF event days.

While NPF studies over the Tibetan Plateau (TP) are limited due to challenges in in-situ observations under extreme meteorological and topographic conditions, previous studies reported a high frequency of NPF events in TP. For instance, Tang et al. (2023) observed frequent NPF (80%) at Nam Co station (4730m a.s.l.) in the central TP during monsoon season. Long-term NPF measurements at the Himalayan Nepal Climate Observatory at Pyramid (NCO-P) site on the southern TP showed very frequent NPF events (~50%) (Venzac et al., 2008). These studies emphasize the significance and climatology of NPF in TP.

To address your concern, we have added detailed descriptions of the NPF events in the Lines 142–143 in Section 2.1 and in the Lines 295–297 Section 3.1 of the revised manuscript. Additionally, we have included a table providing statistics on NPF events in the supplementary information for further clarification (Table R1).

*Comments 3: Page 7: line 146-147- "In this study, the simulation domain covers the southeastern TP, with a grid resolution of 27 km and 30 vertical layers… 100 hPa."*

*What uncertainties are linked to the coarser model resolution? Can a grid resolution of 27 km adequately capture certain cloud structures, sub-regional aerosol and cloud variations, and boundary layer dynamics?*

**Response:** Thank you for your comment. The choice of a 27 km resolution for the simulation domain was made to strike a balance between computational efficiency and the need to capture regional-scale atmospheric processes over the southeastern Tibetan Plateau. We appreciate your concern about potential uncertainties linked to this coarser model resolution.

The intricate terrain of the TP poses challenges for coarse-resolution simulations, particularly in accurately representing orographic effects and fine-scale atmospheric processes. Coarse horizontal resolutions may struggle to capture meso- and micro-scale weather systems closely tied to complex terrain (Lindzen et al., 1983; Wang et al., 2020). Additionally, challenges arise in representing convection, cloud microphysics, and boundary layer dynamics at coarser resolutions (Sato et al., 2008; Jain et al., 2019; Mishra et al., 2018).

As widely acknowledged, accurate simulation of certain cloud structures and prediction of sub-regional dynamics are both challenging for regional/global models and even for convective-permitting models, especially on the TP. Studies have shown that increasing spatial resolution does not always lead to higher accuracy due to uncertainties in physics parameterizations and numerical algorithms (Zhang et al., 2013; Wehner et al., 2014; Bacmeister et al., 2014; Yu et al., 2015; Hart et al., 2005). While the 27 km resolution may not fully capture sub-regional cloud structures and aerosol distribution, it is crucial to emphasize that our primary focus is on regional-scale dynamics. Several studies with similar resolutions have demonstrated the model's ability to reproduce observations within an acceptable bias range (Li et al., 2015; Shi et al., 2008; Gao et al., 2018; Yang et al., 2018; Xu et al., 2018; Rai et al., 2022). Gao et al. (2018) found the improved performance of precipitation simulations over the Tibetan Plateau with the WRF model at around 30 km resolution compared to coarser resolution GCMs and reanalysis. Yang et al. (2018), utilizing the WRF-Chem model at 25 km resolution, demonstrated an appropriate representation of the position of the rain band and near-surface aerosol concentrations.

Although the 27 km and 30 layers resolution introduce limitations in capturing fine-scale boundary layer dynamics, studies (Ding et al., 2016; Huang et al., 2020) have indicated that, for regional-scale simulations, the use of 27 km and 30 layers resolution model on the representation of boundary layer processes is within an acceptable range.

In summary, the choice of a 27 km and 30 layers resolution represents a compromise between computational efficiency and the need to capture key regional-scale atmospheric processes over the southeastern Tibetan Plateau. While uncertainties accompany 27 km and 30 layers resolution, especially in representing fine-scale features, previous studies and the specific goals of our research support the chosen resolution. Considering referee's points, we have incorporated additional discussions to address the concerns of uncertainties linked to model resolution in the revised Section 2.2 of the manuscript (Lines 160–164).

*Comment 4: Page 9, Table 1- The authors mention the MB and RMSE over the entire study period. It's important to understand whether the model captures the diurnal variability of meteorological factors, aerosols, and trace gas components. This is especially critical for accurately reproducing NPF characteristics. While the table describes the comparison at Lulang, and the average values seem promising, is it possible to compare/validate the regional distribution of some of these parameters?*

**Response:** Thanks for the comment. To address your concerns, we have added the

evaluations of our model's performance by incorporating a comparison of the spatial patterns of observed 2-meter temperature, 10-meter wind speed, $SO_2$ concentrations, and $PM_{2.5}$ concentrations with model simulations.

Fig. R1a-d illustrates the comparison of our simulation results with observed meteorological fields obtained from the Integrated Surface Database (https://www.ncei.noaa.gov/products/land-based-station/integrated-surface-database). The model demonstrates a commendable ability to capture the topographically induced contrast in temperature between the Tibetan Plateau and South Asia, as well as accurately representing wind speed patterns.

To further validate the regional distribution of key parameters, we utilized air quality data from Chinese national monitoring stations, sourced from the China National Environmental Monitoring Centre. Specifically, our focus was on evaluating the model's simulation of NPF-relevant pollutants, e.g. $SO_2$ and $PM_{2.5}$. Fig. R1e-h presents the spatial patterns of $SO_2$ and $PM_{2.5}$ concentrations, highlighting the model's effectiveness in capturing the regional distribution of these pollutants.

Considering the referee's points, these additional evaluations of spatial distributions have been incorporated into the revised Section 2.2 (Lines 200–205) of the manuscript and Fig. S1 in the supplementary information.

[Figure]

Fig. R1. Averaged spatial distributions of (a) observed and (b) simulated 2-m temperature during 28–30 April. Averaged spatial distributions of (c) observed and (d) simulated 10-wind speed temperature during 28–30 April. Averaged spatial distribution of (e) observed $SO_2$ at Chinese national air quality monitoring stations and (f) simulated $SO_2$ concentrations during 28–30 April. Averaged spatial distribution of (g) observed $PM_{2.5}$ concentrations at Chinese national air quality monitoring stations and (h) simulated $PM_{2.5}$ concentrations during 28–30 April. Here, the surface weather data are

accessible          at          the          Integrated          Surface          Database
(https://www.ncei.noaa.gov/products/land-based-station/integrated-surface-database).
The air quality dataset at the national monitoring stations is available from the China
National Environmental Monitoring Centre (http://www.pm25.in).

*Comment 5: Page 12, Figure 3- Some of the NPF days showed morning peaks in number concentrations, which were associated with wood burning in a nearby residential cottage. Is this a regular practice? If so, residential burning near the measurement site could regularly influence NPF formation. For instance, on April 30th, it appears that NPF growth is influenced by the introduction of particles from wood burning. Is this phenomenon very local or more regional? If this practice is very localized, can the results be considered as specific to this region and not applicable to a larger area such as southern TP?*

**Response:** Thanks for the comment. The morning peaks in number concentrations during 28-40 April, as illustrated in Fig. 3, are associated with wood burning in single nearby residential cottage. This practice is a regular occurrence during morning hours. It is essential to clarify that these biogenic burning signals are specific to very local emissions originating from the mentioned cottage. To assess the potential impact of this local burning on NPF, we examined black carbon (BC) concentrations measured by The Multi-Angle Absorption Photometer (MAAP, Thermo-Scientific Inc.) at our site. Notably, the BC concentration was extremely high during the wood burning period while relatively low during NPF event in the afternoon (Fig. R2). This observation suggests that the NPF growth process is minimally influenced by wood burning, reinforcing the notion that our results can be considered applicable to the regional scale. Furthermore, the characteristic "banana" shape in the particle number size distribution surface plot, as depicted in Fig. 3, signifies that nucleation and growth processes occur on a regional scale. This "banana" shape is a well-established criterion for identifying regional NPF events (Kulmala et al., 2012). This supports the generalizability of our findings beyond the localized impact of the nearby cottage, affirming the regional significance of NPF event. Meanwhile, the simulation results also support that the NPF event occurred on a regional scale in the southeastern Tibetan Plateau. Considering the referee's points, we have added more discussions in the Lines 297–299 in the revised Section 3.1 of the manuscript and added Fig. R2 in Supplement Information.

[Figure]

Fig. R2. (a) Temporal evolution of particle number size distribution and (b) BC concentrations measured in Lulang during 28–30 April, 2021. All time is in the UTC+8 time zone in this study.

*Comment 6: Page 12, Figure 3- Please elaborate on the start times and end times of all the NPF events.*

**Response:** Thanks for the comment. The range of start times of all NPF events is 9:00–13:30 and the range of end times is 13:00 –20:10. Considering the referee's points, we have added a table showing statistics (including the start times and end times) on NPF events in the supplementary information (Table S1 or Table. R1).

*Comment 7: Page 15, line 316-318- "The potential role of transport is also suggested by Fig. 3a, which … especially on 29 April."*

*Would this also suggest that there has been cluster formation somewhere else, transported, and that the growth is rather observed at the Lulang site?*

**Response:** The particle number size distribution shown in Fig. 3a reveals a high concentration of particles larger than 10 nm and relatively low concentrations of sub-10 nm particles, especially on April 29. It suggests the potential for cluster formation elsewhere followed by transport to the Lulang site. This observation aligns with our hypothesis that the observed NPF event is significantly influenced by regional transport or convective motions. We have modified the discussions in Lines 353-354 and 356-357 in revised manuscript.

*Comment 8: Page 17, line 348-350- "The typical "banana" shape particle number size distribution showing nucleation and subsequent ... a regional scale."*

*This description could probably be added in either the measurement or the results section when describing the characteristics of NPF.*

**Response:** Thanks for the comment. We added this sentence in the results section when describing the characteristics of NPF in the revised Section 3.1 (Lines 295–297) of the manuscript.

*Comment 9: Page 16, Figure 5- It appears that on the 28th of April, there are preexisting aerosols (30 -90 nm) present in comparison to the observations. This raises the question of whether the emission inventories used in the model are consistent with the observations, a task that is challenging to maintain. Consequently, it would be good to consider what kind of uncertainties might arise from this misrepresentation, if any, in the formation of NPF.*

**Response:** Thanks for the comment. In our study, anthropogenic emissions were sourced from the mosaic Asian anthropogenic emission inventory (MIX) for the year 2010, incorporating specific emissions for China from the Multi-resolution Emission Inventory for China (MEIC) for the year 2017. The MIX inventory, designed for 2008 and 2010, amalgamates up-to-date regional emission inventories to support Model Inter-Comparison Study for Asia (MICS-Asia) and Task Force on Hemispheric Transport of Air Pollution (TF HTAP) initiatives (Li et al., 2017b). MEIC, a widely used bottom-up emission inventory model for chemical simulation in China, encompasses approximately 700 anthropogenic sources over China (Zheng et al., 2018).

It is acknowledged that emission estimates from these inventories are subject to uncertainties due to the lack of complete knowledge of human activities and emissions from different sources. Li et al. (2017b) outlined uncertainty range estimates for MIX, indicating relatively small for species dominated by large-scale combustion sources (e.g., $SO_2$, $NO_x$, and $CO_2$) but larger for species mainly emitted by small-scale and scattered sources (e.g., CO, NMVOC, and carbonaceous aerosols). MEIC uncertainty range estimates were also reported (Li et al., 2017a; Zheng et al., 2018). The challenge of anthropogenic emission inventory uncertainties is particularly pronounced in the Tibetan Plateau due to data paucity and unreliable locally measured emission factors.

The mosaic process in compiling MIX introduces uncertainties from

inconsistencies among datasets, missing species, closure of mass balances, and border inconsistencies (Janssens-Maenhout et al., 2015). Moreover, gridded emissions from MIX are only available for 2008 and 2010, which may introduce temporal discrepancies. To partially address this issue, we updated the emissions in China with the MEIC inventory available for the year 2017.

As a result, the uncertainties in emission inventories, especially for NMVOC and carbonaceous aerosols, could influence the simulation of atmospheric oxidation capacity and sinks, potentially affecting the simulation of NPF. The discrepancies between model results and observations shown in Fig. 5 could be attributed to the uncertainties in emission inventories. Considering the referee's points, we have added more discussions to address the uncertainty of the emission inventories in the Section 2.2 (Lines 177–180) of the revised manuscript.

*Comment 10: Page 17, Figure 6. The model results indicate that the nucleation and growth of aerosol particles extended widely over southeastern TP. It is interesting to note that nucleation events are not observed in the northeastern part of India (below southern TP), despite its high impact from biogenic sources, ammonia from agricultural activities, and anthropogenic sources. What are the other underlying conditions that favor nucleation events more in the southern TP region? Can this be explained by the higher OH radical concentration, which is a key oxidant for forming H2SO4, observed in the southeastern TP?*

**Response:** Thanks for the comment. The extended nucleation and growth of aerosol particles over the southeastern Tibetan Plateau (TP), as indicated in Figure 6, indeed raises an interesting contrast with the lack of nucleation events in the northeastern part of India. Previous observations in the northeastern part of India (e.g. in Ranichauri and Mukteshwar) also reported infrequent new particle formation event compared to the high altitudes of TP (Venzac et al., 2008; Neitola et al., 2011; Sebastian et al., 2022). This discrepancy can be attributed to several underlying conditions.

One notable factor is the higher concentration of the hydroxyl radical (OH), a key oxidant crucial for forming sulfuric acid ($H_2SO_4$) and indicative of atmospheric oxidizing capacity. The southeastern TP exhibits elevated OH radical concentrations compared to the northeastern India, creating more favorable conditions for nucleation events. Additionally, the Tibetan Plateau, being a relatively pristine environment, experiences a significantly lower condensation sink (CS) in comparison to South Asia, as illustrated in Fig. 7c. The condensation sink quantifies the role of pre-existing particles in removing condensable vapors and newly formed particles from the atmosphere. The combination of a strong atmospheric oxidizing capacity and a low condensation sink over the Tibetan Plateau creates conditions that

are more conducive to NPF. This sheds light on the intricate interplay of atmospheric factors influencing NPF in different regions.

*Comment 11: Page 19, line 374-377- "In contrast, the OH radical concentration, which is a key oxidant for forming…ozone background over the TP".*

*The OH radical concentrations are also high over the Brahmaputra valley region and Bangladesh (Figure 7b), which are also the regions showing higher aerosols. How does the mechanism of OH formation differ over these regions from southern TP.*

**Response:** Thank you for the comment. The concentration of hydroxyl radicals (OH) is influenced by both sources and sinks. On one hand, OH is a major atmospheric oxidant primarily produced in the global troposphere through the UV photolysis of ozone (Levy, 1971). This process involves the production of electronically excited O(1D) atoms, which react with water molecules to form OH. Therefore, factors such as ozone concentration, air humidity, and UV radiation intensity determine the rate of OH generation. In the polluted lower atmosphere, OH can also be efficiently produced by the photolysis of nitrous acid (HONO) (Fu et al., 2019). It's noteworthy that hydroperoxy radicals ($HO_2$) can react with NO to recycle back to OH, with the photolysis of formaldehyde (HCHO) serving as the primary source of $HO_2$ (Tan et al., 2019; Ehhalt and Rohrer, 2000). On the other hand, reactions with carbon monoxide (CO) and hydrocarbons constitute the major sinks for OH.

In the polluted Brahmaputra valley region and Bangladesh, as depicted in Figs. R4a and R4b, elevated levels of formaldehyde (HCHO) and nitrogen dioxide ($NO_2$) contribute to OH production. HCHO is a precursor for the production of hydroperoxy radicals ($HO_2$), which can further convert to OH. The high concentration of NO in these regions enhances the conversion of $HO_2$ to OH. Furthermore, the UV photolysis of ozone is promoted due to the presence of abundant ozone precursors in these polluted areas (Fig. R4c). The photolysis of HONO also contributes to OH production in the polluted Brahmaputra valley (Fig. R4e). However, it may be underestimated due to the absence of the heterogeneous generation and primary emission of HONO in the current version of model.

In the southern Tibetan Plateau (TP), OH production is facilitated by factors such as intense solar radiation, a high ozone background (Fig. R4c). The presence of abundant water vapor enhances the chemical production of OH radicals over the TP (Fig. R4d). Notably, the lower concentration of alkenes (Fig. R4f) indicates a reduced sink for OH, contributing to the higher OH concentration over the TP. We have more discussions on spatial distribution

of OH in the revised manuscript (Lines 417–419) and present Fig. R3 in the Supplement.

[Figure]

Fig. R3. Average spatial distributions of simulated near-surface (a) formaldehyde (HCHO) concentrations, (b) NO concentrations, (c) $O_3 \times O_3$ photolysis rate, (d) relative humidity, (e) nitrous acid (HONO) concentrations, (f) alkenes concentrations during 28–30 April. Note: The black star marks the location of the Lulang site.

*Comment 12: Page 19, line 391-393- "The vertical simulation of zonal averaged wind vector and NH3 concentrations over 94–96 °E … TP."*

*This is where the coarse resolution of the model could result in the misrepresentation of vertical velocities, which is always difficult to simulate. The Himalayan ranges have complex topography. How do you think this would impact the long-range transportation of pollutants from South Asia? For example, in Figure 8b, the transfer of pollutants across higher elevations could require both local and synoptic influences in terms of advection and vertical mixing. What do the vertical velocities look like in the model simulation?*

**Response:** Thank you for your comment. The impact of the complex topography of the Himalayas on the long-range transportation of pollutants from South Asia to the Tibetan Plateau (TP) is a crucial aspect to consider. The intricate interplay between large-scale circulation and local valley wind patterns significantly influences the transport mechanisms.

Although previous studies have confirmed the transport of pollutants across the Himalayas, the complex topography of the TP complicates transport mechanisms. On one hand, Cao et al. (2010) revealed that the Himalayas acted as a huge barrier to the transport of a large amount of BC over the plateau based on model simulations. On the other hand, some studies found that the valleys across the Himalayas served as channels for the efficient transport of pollutants (Hindman and Upadhyay, 2002; Marinoni et al., 2010).

The complex topography tends to weaken overall wind speed due to orographic drags. However, it simultaneously generates more small-scale mountain-valley circulations and enhances valley winds over the Himalayan region compared to smoother topography. Studies investigating the impact of resolution on transport processes over the TP have primarily focused on moisture using the Weather Research and Forecasting (WRF) model (Shi et al., 2008; Karki et al., 2017; Lin et al., 2018; Wang et al., 2020). These studies found that while the resolved complex topography yielded more small-scale mountain-valley circulations and enhanced valley winds, the overall moisture transport across the Himalayas towards the TP was weaker with complex topography due to orographic drags. In the contrast, Zhang et al. (2020) observed a 50% higher overall transport flux of pollutants across the Himalayas in the simulation with complex topography, even though the overall wind speed was weakened. This was primarily attributed to the strengthened efficiency of near-surface meridional transport towards the TP, the enhanced wind speed at some valleys, and the deeper valley channels associated with the complex topography.

Figs. 8a-b show that high concentrations of $NH_3$ extending from South Asia accumulate on the southern slope of the Himalayas before overflowing into the TP region. To provide further insight into the transport pathways across the mountains, Fig. R4 illustrates the zonal-averaged vertical velocity over 94–96 °E during the daytime. Near the southern part of the mountain, elevated concentrations of $NH_3$ accumulate, reaching heights of up to 5 km, which could be advected to the TP with the prominent southerlies (Fig. R5). Additionally, up-valley winds during the daytime favour the transport of pollutants towards the TP (Fig. R4).

Considering the referee's comments, we have added more discussions in the section 3.2 (Lines 429–434) and supplementary information in the revised

manuscript.

[Figure]

Fig. R4. Latitude–height cross sections of vertical velocity averaged along 94–96°E during the daytime (12:00–18:00 UTC+8) of 28–30 April. Note: The pink dashed lines show the location of the research domain marked in Fig.6a.

*Comment 13: Page 20, Figure 8- To understand the long-range transportation of aerosols, it would be helpful to visualize the spatial distribution of wind vectors at around 3000m.*

**Response:** Thank you for your comment. Fig. R5 illustrates the spatial distribution of wind vectors at around 3000 m together with $NH_3$ concentration. This visualization provides additional insights into the long-range transportation of pollutants, showing that the high concentration of pollutants could be advected to the Tibetan Plateau (TP) via southwesterly winds. In conjunction with the up-valley winds shown in Fig. R4, it supports to illustrate the effective transport of pollutants towards the TP.

Considering the referee's comments, we have added discussion in the section 3.2 (Lines 429–434) and supplementary information in the revised manuscript.

[Figure]

Fig. R5. Spatial distributions of simulated $NH_3$ concentration together with wind fields at 3000 meters averaged from 12:00 to 18:00 during 28–30 April.

*Comment 14: Page 21, Figure 9: The overall contribution looks very informative. However, is it possible to examine the resultant diurnal evolution of the number of aerosol number concentrations measured vs. modeled, like Figure 5c but in terms of diurnal evolution?*

**Response:** Thank you for your comment. Fig. R6 illustrates the comparison between the observed and simulated $CN_{1-40}$. As depicted in the Fig. R6, the overall diurnal pattern is well captured by the simulation, despite a tendency to overestimate particle number concentrations, which could be attributed to various factors, including uncertainties in emission inventories, topography, and model resolution. The evening peak in simulated $CN_{1-40}$ is primarily influenced by the condensation of condensable gases, with potential uncertainties arising from anthropogenic emission inventory limitations, given the scarcity of reliable data and locally measured emission factors.

It's noteworthy that WRF-Chem demonstrates good agreement in reproducing the diurnal variation and magnitude of particle number concentrations during the daytime, which aligns with the primary focus of this study. This lends overall reliability to the results.

Considering the referee's comments, we have added discussions in the section 3.3 in the revised manuscript (Lines 461–463) and Fig. R6 in Supplement Information.

[Figure]

Fig. R6. Diurnal evolutions of observed and simulated 1-40 nm particle number concentrations averaged during 28–30 April over the research domain.

*Comment 15: Page 24, line 460-462- "The lower PM2.5 concentration at the boundary layer top suggests a lower condensation sink ... particles (Fig. S1b).*

*Is the vertical distribution of PM2.5 simulated at Lulang, or is it the spatial average?*

**Response:** Thanks for your comment. The vertical distribution of $PM_{2.5}$ depicted in Fig. S1b represents the averaged simulation results over the region outlined by the pink dashed rectangle in Fig. 6a during the period from 28 to 30 April.

As illustrated in Fig. 6a, our model indicates that the nucleation and growth of aerosol particles extend across a large regional expanse, spanning several thousand kilometers in the southeastern Tibetan Plateau. To provide a more comprehensive understanding of the nucleation and growth processes in this region, our analysis extends beyond Lulang to encompass the broader area marked by the pink dashed rectangle in Fig. 6a.

*Comment 16: Page 24, Figure 11b-c- The vertical mixing follows a diurnal cycle and responds to the diurnal variation of temperature, and the PBL. The lower boundary layer traps more pollutants closer to the surface, as observed in the supplementary figure. The PBL plays a significant role in the diurnal pattern of aerosol loading. So, how is this process unique concerning NPF formation? Is vertical mixing influenced by other factors such as convection or precipitation over the study location?*

**Response:** Thanks for your comment. Our investigation reveals that vertical mixing plays a pivotal role in shaping the vertical heterogeneity of new particle formation (NPF). This process significantly influences the chemistry stratification of NPF-relevant pollutants within the planetary boundary layer (PBL) by modulating the transport processes. NPF-relevant factors exhibit a non-uniform distribution vertically, primarily attributed to the evolution of the PBL. This is evident in the decreasing $PM_{2.5}$ concentrations with increasing altitude, acting as a sink for condensable gases and nanoparticles.

Furthermore, particles formed at NPF-preferential level would influence aerosol number concentrations at different altitudes through vertical mixing. As discussed in section 3.2, near-surface nucleation induced by anthropogenic-biogenic interactions is facilitated over the southeastern Tibetan Plateau, resulting in the formation of substantial nanoparticles. These newly formed nanoparticles are then transported upwards through vertical mixing within the PBL (Fig. 11a). During this ascent, they encounter a notable reduction in air temperature and $PM_{2.5}$ concentration in the upper PBL (Figs. S5b-c). These altered conditions create an environment conducive to the condensational growth of small particles (Fig. 11b), leading to an increased concentration of particles within the 10–40 nm size range in the upper air (Fig. 10c). Subsequently, the particles that undergo growth through condensation in the upper air are effectively transported back down to lower altitudes by vertical mixing (Fig. 11c). This process significantly enhances particle concentrations in the size range of 10–40 nm near the surface, as observed in the $CN_{10–40}$ measurements (Fig. 9b).

As NPF mostly occurred in nearly clear-sky days, the vertical mixing influenced by convection or precipitation were not considered in this study.

*Comment 17: Page 25, line 490-491- "On clear-sky days, a high frequency of NPF exceeding 60% was observed."*

*The frequency of NPF occurrence is not clear in the current study. The measurements were conducted during the period from April 4 to May 24, 2021. Please provide details regarding the number of occurrences, start time, end time, etc. If the study is based solely on selected days, please include this clarification in the measurement section.*

**Response:** Thanks for your comments. We have added a table providing statistics on NPF events in the supplementary information, including the number of occurrences, start time, end time, formation rate and growth rate on NPF days.

Table R1 The statistics of NPF parameters in Lulang. Note: the NPF start and end time, formation rate ($J_{1.5 \text{ nm}}$ and $J_{3 \text{ nm}}$) and growth rate ($GR_{<3 \text{ nm}}$, $GR_{3-7 \text{ nm}}$, $GR_{7-15 \text{ nm}}$) are the median values.

| Parameter | Value |
|---|---|
| NPF frequency | 21.3% |
| NPF frequency of clear-sky day | 66.7% |
| NPF start time (UTC+8) | 11:50 |
| NPF end time (UTC+8) | 17:00 |
| $J_{1.5 \text{ nm}}$ ($cm^{-3}$ $s^{-1}$) | 1.3 |
| $J_{3 \text{ nm}}$ ($cm^{-3}$ $s^{-1}$) | 0.2 |
| $GR_{<3 \text{ nm}}$ ($nm$ $h^{-1}$) | 0.6 |
| $GR_{3-7 \text{ nm}}$ ($nm$ $h^{-1}$) | 2.5 |
| $GR_{7-15 \text{ nm}}$ ($nm$ $h^{-1}$) | 5.4 |

**References**

Bacmeister, J. T.; Wehner, M. F.; Neale, R. B.; Gettelman, A.; Hannay, C.; Lauritzen, P. H.; Caron, J. M. , and Truesdale, J. E.: Exploratory High-Resolution Climate Simulations using the Community Atmosphere Model (CAM), J. Clim., 27, 3073-3099, 10.1175/jcli-d-13-00387.1, 2014.

Cai, R.; Chandra, I.; Yang, D.; Yao, L.; Fu, Y.; Li, X.; Lu, Y.; Luo, L.; Hao, J.; Ma, Y., et al.: Estimating the influence of transport on aerosol size distributions during new particle formation events, Atmos. Chem. Phys., 18, 16587-16599, 10.5194/acp-18-16587-2018, 2018.

Cao, J. J.; Tie, X. X.; Xu, B. Q.; Zhao, Z. Z.; Zhu, C. S.; Li, G. H. , and Liu, S. X.: Measuring and modeling black carbon (BC) contamination in the SE Tibetan Plateau, Journal of Atmospheric Chemistry, 67, 45-60, 10.1007/s10874-011-9202-5, 2010.

Dal Maso, M.; Kulmala, M.; Riipinen, I.; Wagner, R.; Hussein, T.; Aalto, P. P. , and Lehtinen, K. E. J.: Formation and growth of fresh atmospheric aerosols: eight years of aerosol size distribution data from SMEAR II, Hyytiala, Finland, Boreal Environ. Res., 10, 323-336, 2005.

Ding, A. J.; Huang, X.; Nie, W.; Sun, J. N.; Kerminen, V. M.; Petäjä, T.; Su, H.; Cheng, Y. F.; Yang, X. Q.; Wang, M. H., et al.: Enhanced haze pollution by black carbon in megacities in China, Geophys. Res. Lett., 43, 2873-2879, 10.1002/2016gl067745, 2016.

Ehhalt, D. H. , and Rohrer, F.: Dependence of the OH concentration on solar UV, J. Geophys. Res.-Atmos., 105, 3565-3571, 10.1029/1999jd901070, 2000.

Fu, X.; Wang, T.; Zhang, L.; Li, Q. Y.; Wang, Z.; Xia, M.; Yun, H.; Wang, W. H.; Yu, C.; Yue, D. L., et al.: The significant contribution of HONO to secondary pollutants during a severe winter pollution event in southern China, Atmos. Chem. Phys., 19, 1-14, 10.5194/acp-19-1-2019, 2019.

Gao, Y. H.; Xiao, L. H.; Chen, D. L.; Xu, J. W. , and Zhang, H. W.: Comparison between past and future extreme precipitations simulated by global and regional climate models over the Tibetan Plateau, Int. J. Climatol., 38, 1285-1297, 10.1002/joc.5243, 2018.

Guenther, A.; Karl, T.; Harley, P.; Wiedinmyer, C.; Palmer, P. I. , and Geron, C.: Estimates of global terrestrial isoprene emissions using MEGAN (Model of Emissions of Gases and Aerosols from Nature), Atmos. Chem. Phys., 6, 3181-3210, 10.5194/acp-6-3181-2006, 2006.

Hart, K. A.; Steenburgh, W. J. , and Onton, D. J.: Model forecast improvements with decreased horizontal grid spacing over finescale intermountain orography during the 2002 Olympic Winter Games, Weather and Forecasting, 20, 558-576, 10.1175/waf865.1, 2005.

Hindman, E. E. , and Upadhyay, B. P.: Air pollution transport in the Himalayas of Nepal and Tibet during the 1995–1996 dry season, Atmos. Environ., 36, 727-739, https://doi.org/10.1016/S1352-2310(01)00495-2, 2002.

Huang, X.; Ding, A.; Wang, Z.; Ding, K.; Gao, J.; Chai, F. , and Fu, C.: Amplified transboundary transport of haze by aerosol–boundary layer interaction in China, Nat. Geosci., 13, 428-434, 10.1038/s41561-020-0583-4, 2020.

Jain, S.; Mishra, S. K.; Salunke, P. , and Sahany, S.: Importance of the resolution of surface topography vis-a-vis atmospheric and surface processes in the simulation of the climate of Himalaya-Tibet highland, Clim. Dyn., 52, 4735-4748, 10.1007/s00382-018-4411-0, 2019.

Janssens-Maenhout, G.; Crippa, M.; Guizzardi, D.; Dentener, F.; Muntean, M.; Pouliot, G.; Keating, T.; Zhang, Q.; Kurokawa, J.; Wankmüller, R., et al.: HTAP_v2.2: a mosaic of regional and global emission grid maps for 2008 and 2010 to study hemispheric transport of air pollution, Atmos. Chem. Phys., 15, 11411-11432, 10.5194/acp-15-11411-2015, 2015.

Karki, R.; ul Hasson, S.; Gerlitz, L.; Schickhoff, U.; Scholten, T. , and Böhner, J.: Quantifying the added value of convection-permitting climate simulations in complex terrain: a systematic evaluation of WRF over the Himalayas, Earth System Dynamics, 8, 507-528, 10.5194/esd-8-507-2017, 2017.

Kulmala, M.; Petaja, T.; Nieminen, T.; Sipila, M.; Manninen, H. E.; Lehtipalo, K.; Dal Maso, M.; Aalto, P. P.; Junninen, H.; Paasonen, P., et al.: Measurement of the nucleation of atmospheric aerosol particles, Nat Protoc, 7, 1651-1667, 10.1038/nprot.2012.091, 2012.

Levy, H.: NORMAL ATMOSPHERE - LARGE RADICAL AND FORMALDEHYDE CONCENTRATIONS PREDICTED, Science, 173, 141-&, 10.1126/science.173.3992.141, 1971.

Li, J.; Yu, R. C.; Yuan, W. H.; Chen, H. M.; Sun, W. , and Zhang, Y.: Precipitation over East Asia simulated by NCAR CAM5 at different horizontal resolutions, Journal of Advances in Modeling Earth Systems, 7, 774-790, 10.1002/2014ms000414, 2015.

Li, M.; Liu, H.; Geng, G. N.; Hong, C. P.; Liu, F.; Song, Y.; Tong, D.; Zheng, B.; Cui, H. Y.; Man, H. Y., et al.: Anthropogenic emission inventories in China: a review, Natl. Sci. Rev., 4, 834-866, 10.1093/nsr/nwx150, 2017a.

Li, M.; Zhang, Q.; Kurokawa, J.; Woo, J. H.; He, K. B.; Lu, Z. F.; Ohara, T.; Song, Y.; Streets, D. G.; Carmichael, G. R., et al.: MIX: a mosaic Asian anthropogenic emission inventory under the international collaboration framework of the MICS-Asia and HTAP, Atmos. Chem. Phys., 17, 935-963, 10.5194/acp-17-935-2017, 2017b.

Lin, C. G.; Chen, D. L.; Yang, K. , and Ou, T. H.: Impact of model resolution on simulating the water vapor transport through the central Himalayas: implication for models' wet bias over the Tibetan Plateau, Clim. Dyn., 51, 3195-3207, 10.1007/s00382-018-4074-x, 2018.

Lindzen, R. S.; Farrell, B. , and Rosenthal, A. J.: ABSOLUTE BAROTROPIC INSTABILITY AND MONSOON DEPRESSIONS, J Atmos Sci, 40, 1178-1184, 10.1175/1520-0469(1983)040<1178:Abiamd>2.0.Co;2, 1983.

Marinoni, A.; Cristofanelli, P.; Laj, P.; Duchi, R.; Calzolari, F.; Decesari, S.; Sellegri, K.; Vuillermoz, E.;

Verza, G. P.; Villani, P., et al.: Aerosol mass and black carbon concentrations, a two year record at NCO-P (5079 m, Southern Himalayas), Atmos. Chem. Phys., 10, 8551-8562, 10.5194/acp-10-8551-2010, 2010.

Mishra, S. K.; Anand, A.; Fasullo, J. , and Bhagat, S.: Importance of the Resolution of Surface Topography in Indian Monsoon Simulation, J. Clim., 31, 4879-4898, 10.1175/jcli-d-17-0324.1, 2018.

Neitola, K.; Asmi, E.; Komppula, M.; Hyvärinen, A. P.; Raatikainen, T.; Panwar, T. S.; Sharma, V. P. , and Lihavainen, H.: New particle formation infrequently observed in Himalayan foothills – why?, Atmos. Chem. Phys., 11, 8447-8458, 10.5194/acp-11-8447-2011, 2011.

Rai, M.; Kang, S. C.; Yang, J. H.; Chen, X. T.; Hu, Y. L. , and Rupakheti, D.: Tracing Atmospheric Anthropogenic Black Carbon and Its Potential Radiative Response Over Pan-Third Pole Region: A Synoptic-Scale Analysis Using WRF-Chem, J. Geophys. Res.-Atmos., 127, 26, 10.1029/2021jd035772, 2022.

Sato, T.; Yoshikane, T.; Satoh, M.; Miltra, H. , and Fujinami, H.: Resolution Dependency of the Diurnal Cycle of Convective Clouds over the Tibetan Plateau in a Mesoscale Model, J. Meteorol. Soc. Jpn., 86A, 17-31, 10.2151/jmsj.86A.17, 2008.

Sebastian, M.; Kompalli, S. K.; Kumar, V. A.; Jose, S.; Babu, S. S.; Pandithurai, G.; Singh, S.; Hooda, R. K.; Soni, V. K.; Pierce, J. R., et al.: Observations of particle number size distributions and new particle formation in six Indian locations, Atmos. Chem. Phys., 22, 4491-4508, 10.5194/acp-22-4491-2022, 2022.

Shi, X. Y.; Wang, Y. Q. , and Xu, X. D.: Effect of mesoscale topography over the Tibetan Plateau on summer precipitation in China: A regional model study, Geophys. Res. Lett., 35, 5, 10.1029/2008gl034740, 2008.

Tan, Z. F.; Lu, K. D.; Hofzumahaus, A.; Fuchs, H.; Bohn, B.; Holland, F.; Liu, Y. H.; Rohrer, F.; Shao, M.; Sun, K., et al.: Experimental budgets of OH, $HO_2$, and $RO_2$ radicals and implications for ozone formation in the Pearl River Delta in China 2014, Atmos. Chem. Phys., 19, 7129-7150, 10.5194/acp-19-7129-2019, 2019.

Tang, L.; Hu, M.; Shang, D.; Fang, X.; Mao, J.; Xu, W.; Zhou, J.; Zhao, W.; Wang, Y.; Zhang, C., et al.: High frequency of new particle formation events driven by summer monsoon in the central Tibetan Plateau, China, Atmos. Chem. Phys., 23, 4343-4359, 10.5194/acp-23-4343-2023, 2023.

Venzac, H.; Sellegri, K.; Laj, P.; Villani, P.; Bonasoni, P.; Marinoni, A.; Cristofanelli, P.; Calzolari, F.; Fuzzi, S.; Decesari, S., et al.: High frequency new particle formation in the Himalayas, Proc. Natl. Acad. Sci. U. S. A., 105, 15666-15671, 10.1073/pnas.0801355105, 2008.

Wang, Y.; Yang, K.; Zhou, X.; Chen, D. L.; Lu, H.; Ouyang, L.; Chen, Y. Y.; Lazhu , and Wang, B. B.: Synergy of orographic drag parameterization and high resolution greatly reduces biases of WRF-simulated precipitation in central Himalaya, Clim. Dyn., 54, 1729-1740, 10.1007/s00382-019-05080-w, 2020.

Wehner, M. F.; Reed, K. A.; Li, F. Y.; Prabhat; Bacmeister, J.; Chen, C. T.; Paciorek, C.; Gleckler, P. J.; Sperber, K. R.; Collins, W. D., et al.: The effect of horizontal resolution on simulation quality in the Community Atmospheric Model, CAM5.1, Journal of Advances in Modeling Earth Systems, 6, 980-997, 10.1002/2013ms000276, 2014.

Xu, J. W.; Koldunov, N.; Remedio, A. R. C.; Sein, D. V.; Zhi, X. F.; Jiang, X.; Xu, M.; Zhu, X. H.; Fraedrich, K. , and Jacob, D.: On the role of horizontal resolution over the Tibetan Plateau in the REMO regional climate model, Clim. Dyn., 51, 4525-4542, 10.1007/s00382-018-4085-7, 2018.

Yang, J. H.; Kang, S. C.; Ji, Z. M. , and Chen, D. L.: Modeling the Origin of Anthropogenic Black Carbon

and Its Climatic Effect Over the Tibetan Plateau and Surrounding Regions, J. Geophys. Res.-Atmos., 123, 671-692, 10.1002/2017jd027282, 2018.

Yu, R. C.; Li, J.; Zhang, Y. , and Chen, H. M.: Improvement of rainfall simulation on the steep edge of the Tibetan Plateau by using a finite-difference transport scheme in CAM5, Clim. Dyn., 45, 2937-2948, 10.1007/s00382-015-2515-3, 2015.

Zhang, H. L.; Pu, Z. X. , and Zhang, X. B.: Examination of Errors in Near-Surface Temperature and Wind from WRF Numerical Simulations in Regions of Complex Terrain, Weather and Forecasting, 28, 893-914, 10.1175/waf-d-12-00109.1, 2013.

Zhang, M. X.; Zhao, C.; Cong, Z. Y.; Du, Q. Y.; Xu, M. Y.; Chen, Y.; Chen, M.; Li, R.; Fu, Y. F.; Zhong, L., et al.: Impact of topography on black carbon transport to the southern Tibetan Plateau during the pre-monsoon season and its climatic implication, Atmos. Chem. Phys., 20, 5923-5943, 10.5194/acp-20-5923-2020, 2020.

Zheng, B.; Tong, D.; Li, M.; Liu, F.; Hong, C.; Geng, G.; Li, H.; Li, X.; Peng, L.; Qi, J., et al.: Trends in China's anthropogenic emissions since 2010 as the consequence of clean air actions, Atmos. Chem. Phys., 18, 14095-14111, 10.5194/acp-18-14095-2018, 2018.

---

## Author Comment (AC2)

*Review of New particle formation induced by anthropogenic-biogenic interactions in the southeastern Tibetan Plateau, Lai et al.*

*This manuscript contains a suite of high-quality measurement data showing H2SO4, HOMs, and a PNSD from the Tibetan plateau. Frequent NPF was observed and an analysis of both the measurement data and some very impressive WRF-Chem simulations are presented giving some fascinating insights into the chemistry on a larger scale than measurements allow. The WRF-Chem simulations are possible due to some improvements to the VBS. The methodology for both the measurements and model are, however, extremely sparse. The developments to VBS are not discussed at all, neither are the processing of the measurement data. Similarly, the analysis of the mass spectral data is not given enough time. Many of the key arguments depend on the HOMs being monoterpene oxidation products, but the mass spectra are not discussed in detail. I understand that this would result in a very large paper so I think these things belong in the supplement. Once these are addressed, I very highly recommend this for publication as it is an extremely impressive paper.*

**Response:** We would like to thank the referee for providing the insightful suggestions, which indeed help us further improve the manuscript.

**General comments**

*1) The methodology regarding instrumentation is a little bit thin-on-the-ground. It would be nice to have more information about the equipment. What flow rates were the instruments run at? Did they share an inlet? Did the PSM run in scanning mode? What were the time resolution? How were the PTR and CIMS instruments calibrated? What about mass-dependent transmission corrections? This information can go in the supplement, but it is important.*

**Response:** Thanks for your comment. The particle number size distributions (PNSDs) in the size range from 1 nm to 20 µm were collectively measured using five instruments, including a Particle Size Magnifier (PSM, Airmodus Inc.), a Neutral cluster and Air Ion Spectrometer (NAIS, Ariel Inc.), two Scanning Mobility Particle Sizers (nano-SMPS and long-SMPS, TSI Inc.) and an Aerodynamic Particle Sizer (APS, TSI Inc.). The PSM was operated in the scanning mode, with the saturator flow rate continuously changing from 0.1 L/min to 1.3 L/min. The NAIS observed air ion number size distributions from 0.8 nm to 40 nm and particle number size distributions from 2 nm to 40 nm. The nano-SMPS and long-SMPS share similar configurations but differ by a differential mobility analyzer (DMA) and a condensation particle counter (CPC) for measuring PNSDs in the size range of 4–70 nm and 12–540 nm, respectively. The APS measures the PNSD from 500 nm to 20 µm in the aerodynamic diameter. The sample air of nano-SMPS, long-SMPS and APS were dried using the silica gel dryer. The inlet flow rate of PSM, NAIS, nano-SMPS, long-SMPS and APS were 2.5 L/min, 54 L/min, 1.5 L/min, 1 L/min and 5 L/min, respectively. Throughout the intensive campaign, nano-SMPS and long-SMPS shared one inlet tube while the other instruments used separated inlet tubes. The PNSD in the overlapping size ranges detected by different particle sizers demonstrated good agreement. To obtain the PNSD from 4 nm to 1000 nm, the SMPS data and APS data were merged by following the method described by Beddows et al. (2010). The time resolution of PSM, NAIS, nano-SMPS, long-SMPS and APS data were 4 min, 3.5 min, 5.5 min, 5.5 min and 5.5 min, respectively.

The monoterpene concentration was measured by a Proton Transfer Reaction Time-

Of-Flight Mass Spectrometer (PTR-TOF-MS, Ionicon Analytik Inc.). During the campaign, the transmission function of the PTR-TOF-MS was calibrated with a 15-component gas mixture standard that included isoprene, α-pinene, benzene, and toluene. A nitrate Chemical Ionization with the Atmospheric Pressure interface Time-Of-Flight mass spectrometer (CI-APi-TOF, Aerodyne Research Inc.) was used to detect the $H_2SO_4$ and HOMs (Jokinen et al., 2012). $H_2SO_4$ was calibrated during the campaign by utilizing a stable and adjustable concentration $H_2SO_4$ source (Kürten et al., 2012). The mass-dependent transmission correction of the CI-APi-TOF was determined by following the method described by Heinritzi et al. (2016), by depleting the reagent ions with several perfluorinated acids before the campaign.

To address your concern, we have included additional information regarding instrumentation (section S1) in the supplementary information section for further clarity.

2) *Comment: The improvements to WRF-Chem are very valuable! But barely discussed. You say you updated VBS to incorporate RO2 chemistry, including autoxidation and dimerization! Is this similar to existing work such as PRAM/autoPRAM? (1). This should definitely be discussed in detail (again, even if only in the supplement) as the outcomes of the paper hinge on these results. Also, are there plans to make these improvements available widely?*

**Response:** Thanks for your comment.

Previous studies have demonstrated that oxidation of monoterpene could generate large amounts of ultra- and extremely low-volatility organic compounds, which are important precursors for NPF (Ehn et al., 2014). Thus, in our study, the peroxy radicals ($RO_2$) chemistry we accounted for is specific to monoterpenes. The modified volatility basis set (VBS) framework we used in WRF-Chem explicitly represents peroxy radicals ($RO_2$) chemistry and distributes products into the appropriate volatility bins after $RO_2$ termination into stable molecules. In the modified VBS framework, organic species are lumped into the $C^*$ bins, with $C^*$ ranging from $10^{-9}$ to $10^5$ μg m$^{-3}$, separated by powers of $10^2$ (i.e., 8 bins in total).

The reactions begin with oxidation of monoterpene with $O_3$, OH, and $NO_3$, producing peroxy radicals ($RO_2$). The model distinguishes two types of peroxy radicals: one with the potential for autoxidation and the other without. The radical termination proceeds via unimolecular termination or reactions with $HO_2$, NO, or another peroxy radical. Peroxy radical cross-reactions can produce dimers (ROOR), and the fraction of dimers in all cross-reaction products is assumed to depend on the volatility of the reacting peroxy radicals. The non-dimer cross-reaction products, as well as the termination products via unimolecular termination or reaction with NO, undergo either functionalization or fragmentation. The reactions and rate coefficients in our work are summarized in Table S1 of Schervish and Donahue (2020). It is noteworthy that the mechanism of peroxy radical autoxidation used in this study was similar with that in PRAM/autoPRAM (Roldin et al., 2019), but it is a simplified version due to the computational efficiency of the regional transport model. We map the stable molecules generated from each peroxy radical termination pathway to a distribution of species in the VBS space through kernels, allowing us to represent the wide variety of both peroxy radicals and stabilization reactions (Schervish and Donahue, 2020). The kernels used in this work are summarized in Tables S3–S6 of Schervish and Donahue (2020).

To address your comment, we have added discussions in the supplementary information (section S2) to provide more details about the improvements made to the WRF-Chem model, especially regarding the updates to the VBS. Additionally, we want to clarify that the modified codes of WRF-Chem in this study are available

upon request to the corresponding authors.

3) *Similar to the previous comments, as your arguments hinge on the HOMs being monoterpene oxidation products, it would be nice if you showed them in more detail. The mass defect shows them, but you've lumped C6-10 together. Why not colour it by carbon number? I'd also like to see DBE per carbon, and average oxidation state. Otherwise I have no idea what the HOMs actually are. I'd need to be satisfied that the HOMs are similar to alpha-pinene oxidation products, as many of the later arguments depend on alpha pinene lab studies.*

**Response:** Thanks for your comment. We have modified the mass defect plot (Fig. 3c or Fig. R1c) in the revised manuscript, by colouring the values by carbon number. Fig. R1c shows the mass defect plots of negative ion cluster during the NPF period on 29 April. Many ions with m/z values higher than 300 Th and carbon number higher than 10 were observed during the NPF, suggesting the contribution of HOMs to the nucleation.

During 28-30 April, 2021, the median double bond equivalent (DBE) and DBE per carbon of neutral HOMs observed by CI-APi-TOF were 2.97 and 0.34, respectively. Moreover, the C10 HOMs dominated the carbon distribution of HOMs, with the fraction of 32%. All those evidences suggest that the HOMs are mainly formed by the oxidation of monoterpene oxidation. The detailed data analysis of CI-APi-TOF data (e.g. bin-PMF etc) is beyond the scope of this study. Our another work will focus on the pathway of HOMs formation from monoterpene oxidation in the southeastern part of Tibetan Plateau. Nevertheless, we have presented required results from CI-APi-TOF measurements to support that HOMs in the southeastern part of Tibetan Plateau are mainly from monoterpene oxidation in the revised manuscript (Lines 327-332).

We indeed improved the WRF-Chem model based on the mechanism of alpha-pinene oxidation, as a large number of lab experiments have thoroughly investigated this mechanism. Various monoterpenes including alpha-pinene, beta-pinene and limonene are grouped together in the WRF-Chem, which leads to the uncertainties in the modelling of HOMs. The yield of HOMs is higher for alpha-pinene than for beta-pinene, while lower than for limonene. In the revised manuscript, we have added the discussions on the model uncertainties in the supplementary information (section S2).

[Figure]

Fig. R1. (a) Nucleation rates ($J_{1.7}$) as a function of $H_2SO_4$ concentration at ambient observations in Lulang (green circles), Hyytiälä (gray circles) (Sihto et al., 2006; Kulmala et al., 2013) and CLOUD experiments (red diamonds) (Lehtipalo et al., 2018). The cyan and blue lines denote ternary ($H_2SO_4$-$NH_3$-$H_2O$) nucleation and binary nucleation ($H_2SO_4$- $H_2O$), respectively, based on CLOUD data in Kürten et al. (2016). (b) Averaged diurnal variations of $H_2SO_4$ concentrations and HOMs concentration on NPF days in Lulang. The solid lines are the median values and shaded areas denote the 25[th] or 75[th] percentiles. (c) A mass defect plot illustrating the chemical composition of negative ion clusters at 12:00 on 29 April. The size and color of symbol size correspond to the relative signal intensity on a logarithmic scale and carbon number, respectively. (d) Formation rate at 10 nm ($J_{10}$) versus formation rate at 3 nm ($J_3$) at ambient observations in Lulang (diamonds). Diamonds are color-coded by condensation sink. Error bars present the 25[th] - 75[th] percentiles. The solid grey line shows the relationship between $J_{10}$ and $J_3$ based on theory (Kulmala et al., 2012) and the uncertainties are shown by the shaded bands. Dash 1:1 line is shown for reference.

**Specific comments**

*1. Line 223: I'm not sure I understand the logic here, why only those species with $C^*$ of $10^{-9}$ and $10^{-7}$? Why not $10^{-8}$? Surely this would avoid you having to input the factor-of-six adjustment?*

**Response:** Thanks for your comment.

In this study, we aimed to encompass the broad volatility spectrum of organic vapors while balancing computational efficiency. To achieve this, biogenic organic species are lumped into eight volatility bins having saturation vapor concentrations

(C*) of $10^{-10}$–$10^{-9}$, $10^{-8}$–$10^{-7}$, $10^{-6}$–$10^{-5}$, $10^{-4}$–$10^{-3}$, $10^{-2}$–$10^{-1}$, $10^{0}$–$10^{1}$, $10^{2}$–$10^{3}$, $10^{4}$–$10^{5}$ $\mu gm^{-3}$.

Lehtipalo et al. (2018) demonstrated that, among Highly Oxygenated Molecules (HOMs), only organic vapors with ultra-low or extremely low volatility are significant contributors to nucleation. Hence, in our model, surrogate species with C* values lower than $10^{-7}$ $\mu gm^{-3}$ formed from monoterpene oxidation are considered as the nucleating vapors. Considering the referee's points, we clarified in the revised manuscript (Lines 249, 252).

*2. Line 229: Can you explain the temperature dependence function?*

**Response:** Thanks for your comment.

The temperature dependence function for organic nucleation in our study is derived from the work of Dunne et al. (2016).

Dunne et al. (2016) determined the temperature dependence through the Atmospheric Cluster Dynamics Code model (ACDC) studies based on quantum chemical calculations of cluster binding energies. They used the organic proxy compound 3-methyl-1,2,3-butane-tricarboxylic acid (MBTCA) for their calculations. MBTCA was chosen because it is a well-known compound formed in the oxidation of volatile organic compounds and has a high O:C ratio. The formation free energies for MBTCA-sulfuric acid clusters were already available from previous works (Riccobono et al., 2014).

However, it's important to note that this estimation could potentially lead to a stronger temperature dependence than reality. This is because the isomerization reactions that create organic molecules with sufficiently low volatility to participate in nucleation are slower at low temperatures. In these cases, instead of isomerization, organic peroxy radicals react with other peroxy radicals to create stable, less oxidized species, leading to a decrease in oxidation levels.

To address this, Dunne proposed a plausible weaker temperature dependence, which lies between the extremes of zero temperature dependence and the MBTCA case. The expression used for the temperature dependence is as follows:

$$Jorg' = Jorg \exp(-(T - 278)/10)$$

This formula represents the temperature dependence we apply in our model.

To address your concern, we have included additional information regarding the temperature dependence in the supplementary information (section S3) for further clarity.

*3. Line 279: Not sure Qi and Riccobono are the right references here. Maybe these two: (2, 3) as the former shows the formation of particles primarily through HOMs, while the latter shows the importance of HOMs as well as H2SO4 + NH3 in the boreal environment a little more accurately than Riccobono. Maybe also (4) to show their role in growth.*

**Response:** Thanks for your comment. We have modified the references in the revised manuscript (Lines 313–314).

*4. Figure 3: This figure is great. Is it possible to include one for the whole campaign including H2SO4 and HOM? Also maybe use a different colour palette other than Jet (maybe Turbo or Viridis). Same for the other figures.*

**Response:** Thanks for your comment. We modified the Fig. 3a, by using Turbo colormap in the revised manuscript (Fig. R2). The time series of particle number size distribution from 1 nm to 1000 nm, ion number size distribution, $SO_2$, monoterpene, $O_3$, $H_2SO_4$ and HOMs during the whole campaign were presented in Supplemental Information (Fig. R3) and we added descriptions in the revised manuscript (Lines 291–292).

[Figure]

Fig. R2 Temporal evolution of (a) particle number size distribution from 1 nm to 1000 nm, (b) positive ion number size distribution from 0.8 nm to 40 nm. Note that the high peaks of number concentrations around 10–200 nm observed in the morning was caused by the wood burning in a residential cottage near the site. (c) $SO_2$, monoterpene and $O_3$ concentrations measured in Lulang site during 28–30 April, 2021. All time is in the UTC+8 time zone in this study.

[Figure]

Fig. R3. Time series of (a) particle number size distribution from 1 nm to 1000 nm, (b) positive ion number size distribution from 0.8 nm to 40 nm, (c) SO₂, monoterpene and O₃ concentrations, (d) sulfuric acid and HOMs concentrations during the observation campaign in Lulang from 4 April to 24 May, 2021.

*5. Figure 4c: As above, the sequence of greens is quite difficult to understand here. Also, is the choice of red + green for sulphuric acid + H2SO4 color blind friendly?*

**Response:** Thanks for your comment. We have modified Fig. 4c considering color blind friendly (see Fig. R1).

*6. Figure 5b,c: I'm finding these bar charts slightly hard to read. Why do the charts start at <10¹ cm⁻³? It makes the actual difference quite hard to see. Why not a boxplot with a Y axis? Then we'd be able to see the min/max concentrations measured & predicted by the model, as well as the distribution, and median value.*

**Response:** Thank you for the comment. We re-plotted Figure 5b,c using boxplots (see Fig. R4).

[Figure]

**Fig. R4.** (a) Simulated particle number size distribution in Lulang. (b) Observed and simulated H₂SO₄ and monoterpene concentration in Lulang. Bars are the median value during 28–30 April. The horizontal lines of box represent the 66th percentile, median and 33rd percentile and the whiskers represent the 75th and 25th percentiles. (c) Same as Fig. 5b but for number concentrations of 1–10 nm and 10–40 nm particles. (d) The observed and simulated particle number size distributions averaged from 12:00 to 18:00 during 28–30 April. The gray squares and gray circles show the measurements by NAIS and SMPS, respectively. Red, blue and orange lines represent the NPF-on, NPF_inorg and NPF-off experiments.

*7. Figure 5d: It looks like there's a factor of 2-3 difference between the point where the NAIS and SMPS cross over. Do you have a reference instrument you can correct to? If not, it's common practice to correct the NanoSMPS/NAIS to the LongSMPS. In either case, it doesn't make any difference to the conclusions of the figure.*

**Response:** Thank you for the comment. We corrected particle number size distribution observed by NAIS to the that observed by LongSMPS and re-plotted Fig. 5d (see Fig. R4).

*8. Line 384: Do you mean "high values" rather than "certain values"?*

**Response:** Thanks for your comment. Due to the strong atmospheric oxidizing capacity and low condensation sink (CS), H₂SO₄ concentrations can reach relatively high values, especially on the Tibetan Plateau (TP) compared to the surrounding areas. However, it's important to note that, in an absolute sense (or compared with the H₂SO₄ concentration in polluted environments), we do not consider the sulfuric acid concentration to be very high, as the average value is below 1 ppt.

Considering the referee's comments, we have modified the sentence in the revised manuscript (Lines 423–424).

*9. Line 386 (and following paragraph): What about the SO2? If that is also anthropogenic (which I'd presume it is as I doubt there's much DMS up there) then this strengthens your biogenic-anthropogenic argument.*

**Response:** Thanks for your comment. Indeed, $SO_2$ is also of anthropogenic origin and aligns with $NH_3$ in its characteristics in the simulation. $SO_2$, primarily derived from coal combustion in power generation and industrial production, displays high concentrations in South Asia (as depicted in Fig. 7a). In addition to $NH_3$, the southerly winds during the daytime can also transport $SO_2$ to the TP, contributing to NPF over the TP. This additional information reinforces the biogenic-anthropogenic argument. The roles of DMS in $H_2SO_4$ formation were not considered as DMS was not observed and simulated in this study.

Considering the referee's comments, we have incorporated discussions about $SO_2$ in the revised manuscript (Lines 411–413 and Lines 432–434).

*10. Figure 10: Is nucleation rate here J1.5, J3, or J10? Also, it might be easier to read if instead of "binary" and "ternary" you put H2SO4-H2O and H2SO4-NH3-H2O.*

**Response:** Thank you for your comment. In this study, the nucleation rates presented in Figure 10 are based on parameterizations derived from the CLOUD experiments, specifically reporting nucleation rates at a mobility diameter of 1.7 nm. Therefore, strictly speaking, the nucleation rate in this study refers to $J_{1.7}$. Considering the referee's points, we clarified in Figure 10 that the nucleation rate corresponds to $J_{1.7}$ in the revised manuscript (see Fig. R5). We also modified the "binary" and "ternary" to "$H_2SO_4$-$H_2O$" and "$H_2SO_4$-$NH_3$-$H_2O$" in the revised manuscript.

[Figure]

**Fig, R5.** (a) Averaged diurnal evolution of vertical cross section of simulated nucleation rate ($J_{1.7}$) over the research domain during 28–30 April. Note: The black dashed line shows the planetary boundary layer height (PBLH). (b) The relative contribution of different NPF pathways averaged over the research domain during 28–30 April. (c) Same as Fig. 10a but for 10–40 nm particle number concentrations. (d) Same as Fig. 10a but for 10–40 nm biogenic organic mass concentrations.

*11. Line 461: What fraction of total CN10-40 mass does this Biogenic organic mass comprise?*

**Response:** Thanks for your comment. The biogenic organic mass comprises 17% of the total $CN_{10-40}$ mass in our simulation. The concentration of pre-existing particles is overestimated due to the uncertainties in the emission inventories. Therefore, the model could underestimate the biogenic organic mass in the size range 10-40 nm as the HOMs from biogenic sources could condense on large particles.

*12. Line 462: Maybe worth considering that autoxidation rates also decrease with temperature (5)*

**Response:** Thanks for your comment. Indeed, the temperature-dependent factors influencing the condensation process of organic vapors are complex. While autoxidation rates tend to decrease with temperature, the volatility of organics also decreases at lower temperatures, introducing a competing effect (Stolzenburg et al., 2018).

In our study, we found that changes in volatility play a leading role in the PBL. Consequently, the lower temperature at higher altitudes (Fig. S1c) creates more favorable conditions for the condensation process of organic vapors, leading to the counter-gradient of biogenic organic mass (Fig. 10d) and contributing to the

subsequent growth of nanoparticles.

We have added the discussions on the temperature-dependence of autoxidation rates in the revised manuscript (Lines 506–508).

1. *L. Pichelstorfer et al., Towards a mechanistic description of autoxidation chemistry: from precursors to atmospheric implications. EGUsphere **2023**, 1-30 (2023).*
2. *C. Rose et al., Observations of biogenic ion-induced cluster formation in the atmosphere.*
   *Science Advances **4**, eaar5218.*
3. *K. Lehtipalo et al., Multicomponent new particle formation from sulfuric acid, ammonia, and biogenic vapors. Science Advances **4**, eaau5363.*
4. *C. Mohr et al., Molecular identification of organic vapors driving atmospheric nanoparticle growth. Nature Communications **10** (2019).*
5. *D. Stolzenburg et al., Rapid growth of organic aerosol nanoparticles over a wide tropospheric temperature range. Proceedings of the National Academy of Sciences **115**, 9122-9127 (2018).*

**References**

Beddows, D. C. S.; Dall'osto, M. , and Harrison, R. M.: An Enhanced Procedure for the Merging of Atmospheric Particle Size Distribution Data Measured Using Electrical Mobility and Time-of-Flight Analysers, Aerosol Sci. Technol., 44, 930-938, 10.1080/02786826.2010.502159, 2010.

Dunne, E. M.; Gordon, H.; Kurten, A.; Almeida, J.; Duplissy, J.; Williamson, C.; Ortega, I. K.; Pringle, K. J.; Adamov, A.; Baltensperger, U., et al.: Global atmospheric particle formation from CERN CLOUD measurements, Science, 354, 1119-1124, 10.1126/science.aaf2649, 2016.

Ehn, M.; Thornton, J. A.; Kleist, E.; Sipila, M.; Junninen, H.; Pullinen, I.; Springer, M.; Rubach, F.; Tillmann, R.; Lee, B., et al.: A large source of low-volatility secondary organic aerosol, Nature, 506, 476-479, 10.1038/nature13032, 2014.

Heinritzi, M.; Simon, M.; Steiner, G.; Wagner, A. C.; Kürten, A.; Hansel, A. , and Curtius, J.: Characterization of the mass-dependent transmission efficiency of a CIMS, Atmospheric Measurement Techniques, 9, 1449-1460, 10.5194/amt-9-1449-2016, 2016.

Jokinen, T.; Sipila, M.; Junninen, H.; Ehn, M.; Lonn, G.; Hakala, J.; Petaja, T.; Mauldin, R. L.; Kulmala, M. , and Worsnop, D. R.: Atmospheric sulphuric acid and neutral cluster measurements using CI-APi-TOF, Atmos. Chem. Phys., 12, 4117-4125, 10.5194/acp-12-4117-2012, 2012.

Kulmala, M.; Petaja, T.; Nieminen, T.; Sipila, M.; Manninen, H. E.; Lehtipalo, K.; Dal Maso, M.; Aalto, P. P.; Junninen, H.; Paasonen, P., et al.: Measurement of the nucleation of atmospheric aerosol particles, Nat Protoc, 7, 1651-1667, 10.1038/nprot.2012.091, 2012.

Kulmala, M.; Kontkanen, J.; Junninen, H.; Lehtipalo, K.; Manninen, H. E.; Nieminen, T.; Petaja, T.; Sipila, M.; Schobesberger, S.; Rantala, P., et al.: Direct observations of atmospheric aerosol nucleation, Science, 339, 943-946, 10.1126/science.1227385, 2013.

Kürten, A.; Rondo, L.; Ehrhart, S. , and Curtius, J.: Calibration of a Chemical Ionization Mass Spectrometer for the Measurement of Gaseous Sulfuric Acid, Journal of Physical Chemistry A, 116, 6375-6386, 10.1021/jp212123n, 2012.

Kürten, A.; Bianchi, F.; Almeida, J.; Kupiainen-Määttä, O.; Dunne, E. M.; Duplissy, J.; Williamson, C.; Barmet, P.; Breitenlechner, M.; Dommen, J., et al.: Experimental particle formation rates spanning tropospheric sulfuric acid and ammonia abundances, ion production rates, and temperatures, Journal of Geophysical Research: Atmospheres, 121, 10.1002/2015jd023908, 2016.

Lehtipalo, K.; Yan, C.; Dada, L.; Bianchi, F.; Xiao, M.; Wagner, R.; Stolzenburg, D.; Ahonen, L. R.; Amorim, A.; Baccarini, A., et al.: Multicomponent new particle formation from sulfuric acid, ammonia, and biogenic vapors, Science Advances, 4, 9, 10.1126/sciadv.aau5363, 2018.

Riccobono, F.; Schobesberger, S.; Scott, C. E.; Dommen, J.; Ortega, I. K.; Rondo, L.; Almeida, J.; Amorim, A.; Bianchi, F.; Breitenlechner, M., et al.: Oxidation products of biogenic emissions contribute to nucleation of atmospheric particles, Science, 344, 717-721, 10.1126/science.1243527,

2014.

Roldin, P.; Ehn, M.; Kurten, T.; Olenius, T.; Rissanen, M. P.; Sarnela, N.; Elm, J.; Rantala, P.; Hao, L. Q.; Hyttinen, N., et al.: The role of highly oxygenated organic molecules in the Boreal aerosol-cloud-climate system, Nat. Commun., 10, 15, 10.1038/s41467-019-12338-8, 2019.

Schervish, M. , and Donahue, N. M.: Peroxy radical chemistry and the volatility basis set, Atmos. Chem. Phys., 20, 1183-1199, 10.5194/acp-20-1183-2020, 2020.

Sihto, S. L.; Kulmala, M.; Kerminen, V. M.; Dal Maso, M.; Petäjä, T.; Riipinen, I.; Korhonen, H.; Arnold, F.; Janson, R.; Boy, M., et al.: Atmospheric sulphuric acid and aerosol formation: implications from atmospheric measurements for nucleation and early growth mechanisms, Atmos. Chem. Phys., 6, 4079-4091, 10.5194/acp-6-4079-2006, 2006.

Stolzenburg, D.; Fischer, L.; Vogel, A. L.; Heinritzi, M.; Schervish, M.; Simon, M.; Wagner, A. C.; Dada, L.; Ahonen, L. R.; Amorim, A., et al.: Rapid growth of organic aerosol nanoparticles over a wide tropospheric temperature range, Proc Natl Acad Sci U S A, 115, 9122-9127, 10.1073/pnas.1807604115, 2018.

---

## Author Response (AR2)

**Response to Editor Report**

*Thank you for your careful consideration of the referee comments. After thoroughly reviewing the response document and the revised manuscript and SI, I have determined that nearly all of the referee comments have been sufficiently considered. However, I feel that the manuscript would be improved by adding a few more details in response to reviewer 2 comment 1 ("The methodology regarding instrumentation is a little bit thin-on-the-ground. It would be nice to have more information about the equipment.") and 3 ("as your arguments hinge on the HOMs being monoterpene oxidation products, it would be nice if you showed them in more detail").*

**Response:** Thanks for your suggestions, which help us to further improve the manuscript. The changes in the revised manuscript were highlighted in red color. Below is our point-by-point response to each comment.

*Specifically, please consider adding information on the inlets (particularly important for something like H2SO4 and HOMs) as well as further information on how the MS was alternated between CI-APi-TOF and APi-TOF mode (line 144-145; e.g. how long in each mode?, how frequently was APi-TOF mode used?, etc.). Adding these details to the SI is ok.*

**Response:** Thanks for the comments. The $H_2SO_4$ and HOMs were observed by CI-APi-TOF mass spectrometer. The sample air was drawn into CI-APi-TOF through a stainless-steel tube (100 cm long, 3/4 inch diameter) with a flow rate of 10 L min$^{-1}$. The sample flow was then surrounded by a clean sheath flow (25 L min$^{-1}$) containing nitrate reagent ions in a laminar flow reactor. The nitrate reagent ions in the sheath flow were generated by exposing air containing nitric acid to a photoionizer X-ray (Model L9491, Hamamatsu Inc.). To detect the natural cluster ions, the CI-APi-TOF mode was switched to APi-TOF mode by switching off the sheath flow and photoionizer X-ray and setting the ion voltage and drift voltage in laminar flow reactor to zero. On typical NPF (29 Apirl 2021) day, the mass spectrometer was alternated between CI-APi-TOF mode and APi-TOF mode at a frequency of once per hour from 10:00 to 21:00 (UTC+8), with each mode running 1 hour. We have added these details in the SI.

*Regarding the HOMs being monoterpene oxidation products, while Fig. 4c is improved, it is still difficult to tell the intensity of the relevant HOMs in this type of plot. Additionally, as Fig. 4c is an APi-TOF measurement (or so it seems based on the caption), the HOMs distribution may be biased due to measuring naturally charged ions rather than neutral species. I encourage you to consider including something like an average MS (intensity vs m/z) from the CI-APi-TOF and coloring the sticks by the carbon number. This would more clearly show the intensity of the monoterpene oxidation products.*

**Response:** Thanks for the comment. Indeed, the mass spectrum of natural ions clusters might be different with that of neutral molecules. According to your suggestion, we plotted the average mass spectrum of neutral molecules observed by CI-APi-TOF from 28 April to 30 April 2021. As shown in Fig. R1, the spectrum displayed the patterns of HOM monomer and dimer from monoterpene oxidation. The C5-10 HOMs dominated in terms of both species number and concentration. A certain concentration of HOM dimer was observed as well. It furtherly indicates the HOMs were mostly formed from monoterpene oxidation. We have added this result in the revised manuscript (Line 329-332 and Line 334) and Fig. R1 in the SI.

[Figure]

**Figure R1.** Mass spectrum of neutral molecules, colored by carbon number, observed by CI-APi-TOF during 28-30 April, 2021. Note that the top right subplot is zoomed-in view of m/z 450-600.

*Line 331: "with the fraction of 32%." Please clarify what this is a fraction of. I presume it is of the HOMs ion intensity, but it is unclear.*

**Response:** Thanks for the comment. The fraction is of the HOMs concentration. We have clarified it in the revised manuscript in Line 333.

*As mentioned by the technical check, please ensure the color schemes are color blind friendly.*

**Response:** Thanks for the comment. We have checked the Figures 1, 3, 4 using the Coblis and modified Fig. 4b.

[Figure]

**Figure R2.** (a) Nucleation rates ($J_{1.7}$) as a function of $H_2SO_4$ concentration at ambient observations in Lulang (green circles), Hyytiälä (gray circles) (Sihto et al., 2006; Kulmala et al., 2013) and CLOUD experiments (red diamonds) (Lehtipalo et al., 2018). The cyan and blue lines denote ternary ($H_2SO_4$-$NH_3$-$H_2O$) nucleation and binary nucleation ($H_2SO_4$- $H_2O$), respectively, based on CLOUD data in Kürten et al. (2016). (b) Averaged diurnal variations of $H_2SO_4$ concentrations and HOMs concentration on NPF days in Lulang. The lines are the median values and shaded areas denote the 25[th] or 75[th] percentiles. (c) A mass defect plot illustrating the chemical composition of negative ion clusters at 12:00 (UTC+8) on 29 April. The size and color of symbol size correspond to the relative signal intensity on a logarithmic scale and carbon number, respectively. (d) Formation rate at 10 nm ($J_{10}$) versus formation rate at 3 nm ($J_3$) in Lulang. Diamonds are color-coded by condensation sink. Error bars present the 25[th] - 75[th] percentiles. The solid grey line shows the relationship between $J_{10}$ and $J_3$ based on theory (Kulmala et al., 2012) and the uncertainties are shown by the shaded bands. Dash 1:1 line is shown for reference.